# Intercomparison of biogenic $CO_2$ flux models in four urban parks in the city of Zurich

Stavros Stagakis[1], Dominik Brunner[2], Junwei Li[3], Leif Backman[4], Anni Karvonen[5], Lionel Constantin[2], Leena Järvi[5], Minttu Havu[5], Jia Chen[3], Sophie Emberger[6], Liisa Kulmala[4]

[1]Department of Environmental Sciences, University of Basel, Basel, 4056, Switzerland
[2]Empa, Swiss Federal Laboratories for Materials Science and Technology, Dübendorf, 8600, Switzerland
[3]Environmental Sensing and Modeling, Technical University of Munich (TUM), Munich, 80333, Germany
[4]Climate System Research, Finnish Meteorological Institute, Helsinki, FI-00101, Finland
[5]Institute for Atmospheric and Earth System Research (INAR), Physics, University of Helsinki, Helsinki, 00014, Finland
[6]Department of Environmental Systems Science, Institute of Agricultural Sciences, ETH Zurich, Zurich, 8092, Switzerland

*Correspondence to*: Stavros Stagakis (stavros.stagakis@unibas.ch)

**Abstract.** Quantifying the capacity and dynamics of urban carbon dioxide ($CO_2$) emissions and carbon sequestration is becoming increasingly relevant in the development of integrated monitoring systems for urban greenhouse gas emissions. There are multiple challenges towards these goals, such as the partitioning of atmospheric measurements of $CO_2$ fluxes to anthropogenic and biospheric processes, the insufficient understanding of urban biospheric processes, and the applicability of existing biosphere models to urban systems. In this study we applied four biosphere models of varying complexity (diFUME, JSBACH, SUEWS, VPRM) in four urban parks in the city of Zurich and evaluated their performance against in-situ measurements collected over almost two years on park trees and lawns. In addition, we performed an uncertainty analysis of gross primary productivity (GPP), ecosystem respiration ($R_{eco}$), and net ecosystem exchange (NEE) of $CO_2$ based on the differences between the estimates of the four models and compared the estimated uncertainties and biospheric fluxes with the monthly anthropogenic $CO_2$ emissions of a wide urban area surrounding the four parks. The results showed that despite the large differences in model architecture, there was considerable agreement in the seasonal and diurnal GPP, $R_{eco}$ and NEE estimates. Larger discrepancies between the four models were found for lawn GPP compared to tree GPP, while for $R_{eco}$ the differences between lawns and tree areas were similar. On an annual scale, all models agreed, on average, that lawns acted as $CO_2$ sources and tree-covered areas as $CO_2$ sinks during the simulation period, with the exception of diFUME which simulated both tree and lawn areas as $CO_2$ sources. diFUME and VPRM were more accurate in capturing the onset of the tree leaf growth in spring compared to JSBACH and SUEWS. On the other hand, JSBACH and SUEWS simulated soil water availability more accurately than the satellite-derived water index used by VPRM. The in-situ observations revealed a very high spatial variability in lawn $R_{eco}$ across the park areas. All models underestimated the lawn $R_{eco}$ during spring in sunny mowed locations, whereas the model simulations were closer to the observed $R_{eco}$ at un-mowed, partially shaded locations. The mean monthly uncertainties of biogenic NEE reached 0.8 μmol m$^{-2}$ s$^{-1}$, which is 10.2 % of the magnitude of the total $CO_2$ balance over the studied area during the month of June. This balance was composed of a mean anthropogenic

flux of 8.7 μmol m$^{-2}$ s$^{-1}$ and a mean biospheric flux of -0.5 μmol m$^{-2}$ s$^{-1}$. Overall, this study highlights the importance of properly accounting for the biogenic $CO_2$ fluxes and their uncertainties in urban $CO_2$ balance studies, especially during the vegetation growing season, and shows that even simple models, such as VPRM, can adequately simulate the urban biospheric fluxes when appropriately parameterized.

## 1 Introduction

In the battle against the climate crisis, numerous cities have announced ambitious goals to achieve carbon neutrality in the coming decades. Their climate action plans include steps on reducing fossil fuel $CO_2$ emissions and increasing carbon removals by urban green areas. These plans have raised the need for observation-based monitoring of urban greenhouse gas (GHG) fluxes in order to follow the progress towards the climate goals. However, the challenge lies in separating the $CO_2$ fluxes from anthropogenic emissions and biospheric emissions/uptake as their signals are rapidly mixed in the atmosphere (Lauvaux et al., 2020). This is further complicated by the fact that both anthropogenic and biogenic $CO_2$ fluxes have distinct diurnal and seasonal cycles and respond to changes in the environment (Järvi et al., 2019; Stagakis et al., 2023a).

The net $CO_2$ balance or net ecosystem exchange (NEE) of biogenic $CO_2$ fluxes is the difference between ecosystem respiration ($R_{eco}$) and gross photosynthetic $CO_2$ uptake (or gross primary productivity: GPP). NEE is relatively small on an annual basis, especially in urban areas where a big fraction of the land is covered by built structures, but varies significantly on diurnal and seasonal scales (Stagakis et al., 2023b, a; Winbourne et al., 2022) and can significantly alter the atmospheric $CO_2$ observations. Recent studies have shown that urban NEE can offset between 0 and over 100% of the local anthropogenic $CO_2$ emissions depending on the area considered, the source/sink composition, the season, the day type and hour (Lauvaux et al., 2020; Stagakis et al., 2023b, a; Winbourne et al., 2022). A few studies used observations of isotopes and co-emitted species to discriminate between biogenic and anthropogenic $CO_2$ in the urban atmosphere (Pataki et al., 2003; Wu et al., 2022), but the estimates of biogenic $CO_2$ fluxes are associated with high relative uncertainties. Field observations, on the other hand, can provide very valuable information on urban biospheric processes, but are hampered by the extreme heterogeneity of urban environments in terms of vegetation types and species, soil composition, management practices, and local meteorology, as well as technical and logistical restrictions (Lal and Augustin, 2012). This creates a need to utilize models of biogenic $CO_2$ exchange to quantify and further partition the net flux from anthropogenic activities e.g. via inversion methods (e.g. Lauvaux et al., 2020; Stagakis et al., 2023b).

In the past, numerous comprehensive models have focused on carbon exchange processes including photosynthesis, and soil and vegetation respiration at ecosystem level (Hari et al., 2017; Mäkelä et al., 2004; Zhao et al., 2016), but these models were primarily designed for forested or other natural ecosystems. It still remains a question how well they are able to represent carbon uptake processes in urban areas due to the very specific conditions including high variability in soil and vegetation types and species, possible limitations in soil water availability, increased air pollutant loads, higher evaporative demand resulting from elevated temperatures and vapour pressure deficits, special light environments and intensive

management commonly taking place in urban green areas (Dahlhausen et al., 2018; Decina et al., 2016; Nielsen et al., 2007; Wohlfahrt et al., 2019). In recent years, different modelling approaches customized for urban areas have emerged with different strengths and weaknesses. Full carbon cycling/ecosystem models, such as JSBACH (Trémeau et al., 2024), can simulate in detail the processes behind carbon exchanges and carbon stocks. Their downside is that they are commonly aimed to simulate natural ecosystems, they need initial information on soil carbon stocks, and they typically require detailed meteorological observations including precipitation, which are not always available for the urban area of interest. There also exists a set of urban land surface or ecosystem models which can reproduce the urban circumference but do not solve the whole carbon cycle in great detail (SUEWS, Järvi et al., 2019; SURFEX, Goret et al., 2019). What is common to ecosystem and urban models is that they require several parameters and meteorological inputs to be run. Then there exists the group of empirical and light-use-efficiency models that capture the dynamics of vegetation and soil water availability based on satellite observations such as VPRM (Vegetation Photosynthesis and Respiration Model, Mahadevan et al., 2008) and its urban version Urban-VPRM (Hardiman et al., 2017; Sargent et al., 2018), diFUME (Stagakis et al., 2023a) and SMUrF (Wu et al., 2021). They rely on simple ecosystem-specific parameterisations and require only basic meteorological and remote sensing inputs, but they might not be able to catch annual balances with high accuracy as there is no explicit representation of carbon stocks. At the same time, the satellite-based models are tied to present conditions and cannot be used to predict changes in the carbon sinks nor for scenario runs (Havu et al., 2024).

The above models have been already used in simulating urban $CO_2$ fluxes but no systematic comparison or evaluation has previously been made. One reason is the sparseness of observations from urban green areas needed for model evaluation and testing (Hutyra et al., 2014; Winbourne et al., 2022). The challenge in generating representative and comprehensive observations of biogenic $CO_2$ fluxes in urban areas stems from the high spatial and temporal variability of source and sink dynamics, as well as the increased mixing between anthropogenic and biogenic sources (Stagakis et al., 2023b, a). In such heterogeneous environments, discriminating the biogenic $CO_2$ fluxes from the anthropogenic sources is very challenging with conventional approaches such as eddy covariance, especially when considering the observation uncertainties compared to the relative magnitudes of the respective sources (Wu et al., 2022). A number of studies utilizing the eddy covariance technique in urban areas exist (Davis et al., 2017; Järvi et al., 2012; Peters and McFadden, 2012; Velasco et al., 2016), but these measurements are most often limited to the station surroundings and include also some anthropogenic signal even if located in green areas. The change in vegetation and soil carbon pools (e.g., Golubiewski, 2006; Kaye et al., 2005) can be used to derive the uptake on annual or decadal scales, but these are not useful for determining the seasonality and diurnal variability of the fluxes or short-scale responses to drought conditions or heatwaves. Thus, detailed site-specific observations are needed to capture the urban ecosystem dynamics. Direct observations of biogenic $CO_2$ fluxes in urban environments have primarily been obtained through flux chamber measurements, but these can only be applied to certain small scale biospheric features such as soils, low vegetation or individual leaves. Flux chamber measurements have been performed in urban lawns and soils to derive direct observations of lawn NEE, $R_{eco}$ and soil respiration ($R_{soil}$) (Hill et al., 2021; Karvinen et al., 2024; Liss et al., 2009; Weissert et al., 2016). The representativeness of such observations is usually limited due to the large

variability of the soil composition, plant types and management practices across the urban lawns (Lal and Augustin, 2012; Trémeau et al., 2024). Further technical and logistical restrictions to the flux chamber observations arise in urban areas related to private and public spaces, accessibility, safety and aesthetics. Complementary to the flux chamber measurements, key meteorological, physiological and phenological parameters, which act as main environmental drivers or proxies of the biogenic $CO_2$ fluxes, can be measured in-situ. These parameters can help in the interpretation of the measured fluxes, derive

site-specific environmental control functions of $CO_2$ fluxes, or be used directly for the evaluation of model intermediate outputs (Schäfer et al., 2003; Winbourne et al., 2022). Sap flow observations have been used to determine the transpiration of urban trees (Peters et al., 2010; Rahman et al., 2017) and evaluate model performance (Davis et al., 2017; Järvi et al., 2012). The strong coupling between transpiration and photosynthesis can also lead to observation-based estimates of tree GPP (Schäfer et al., 2003). There are however logistical, technical and methodological challenges in measuring sap flow and

deriving representative tree-level transpiration estimates (Ewers and Oren, 2000; Peters et al., 2010). Despite these challenges, several studies have highlighted the value of sap flow measurements to characterize tree physiological responses, especially in diverse and challenging environments where other measurement approaches are not applicable (Peters et al., 2010; Rahman et al., 2017; Schäfer et al., 2003). In conclusion, the number of urban flux observation studies is still very limited and requires better coverage over different vegetation zones and climates.

When simulating the urban biogenic and anthropogenic $CO_2$ fluxes using bottom-up models it is important to provide the corresponding uncertainties. Information on uncertainties is particularly relevant for studies applying inverse methods to estimate anthropogenic and biogenic fluxes across urban areas (e.g. Lauvaux et al., 2016; Stagakis et al., 2023b). In such methods, the relative uncertainties among the inversion inputs can significantly affect the inversion outcomes (Forstmaier et al., 2023; Kunik et al., 2019; Lauvaux et al., 2016; Lian et al., 2022; Wu et al., 2018). It is therefore very important to

provide realistic uncertainty ranges for both anthropogenic and biogenic flux model estimates to avoid cases of overfitting and biases in source attribution. Several studies estimated the uncertainties of ecosystem and surface flux models by performing sensitivity analyses (e.g. Järvi et al., 2019; Stagakis et al., 2023a) or by applying data assimilation approaches (e.g. Santaren et al., 2007; Trotsiuk et al., 2020), but the insights gained from such studies are difficult to be generalised. For example, the sensitivity analysis of the diFUME model when applied in the centre of the city of Basel (Stagakis et al.,

2023a), showed that annual NEE varied between -0.5 and 0.1 kg $CO_2$ m$^{-2}$ a$^{-1}$, which contributed only between -2.8 to 0.5 % to the total $CO_2$ balance when the anthropogenic emissions are taken into account. Furthermore, the uncertainties in the biospheric fluxes and their relative magnitudes to the anthropogenic emissions vary temporally and spatially according to the local characteristics. Since there is still very limited information about biogenic flux model uncertainties in urban areas, these have so far been treated in urban atmospheric inversions in a highly simplified way, e.g. by considering a constant

absolute (Wu et al., 2018) or relative uncertainty (Lian et al., 2022). Several urban flux inversion studies avoided the issue related to biogenic fluxes and their uncertainties by only focusing on the dormant season (Kunik et al., 2019; Lauvaux et al., 2016).

This study applies four biosphere models of varying complexity and sophistication in four urban parks in the city of Zurich where in-situ measurements were collected over almost two years. This work is part of the ICOS-Cities project (ICOS-Cities, 2024), which applies and evaluates the most innovative measurement and modelling approaches of GHG emissions in densely populated urban areas, targeting towards the development of systematic urban GHG observatories in support of cities' climate action planning. The specific objectives of this study are i. to understand and quantify urban biogenic $CO_2$ flux dynamics based on in-situ measurements and model simulations, ii. to evaluate how state-of-the-art biosphere models of different complexity are able to represent these dynamics, iii. to identify the differences, advantages and disadvantages between different model types, and iv. to assess the relative magnitudes of the biogenic $CO_2$ fluxes and their uncertainties as compared to the anthropogenic $CO_2$ emissions in an urban area.

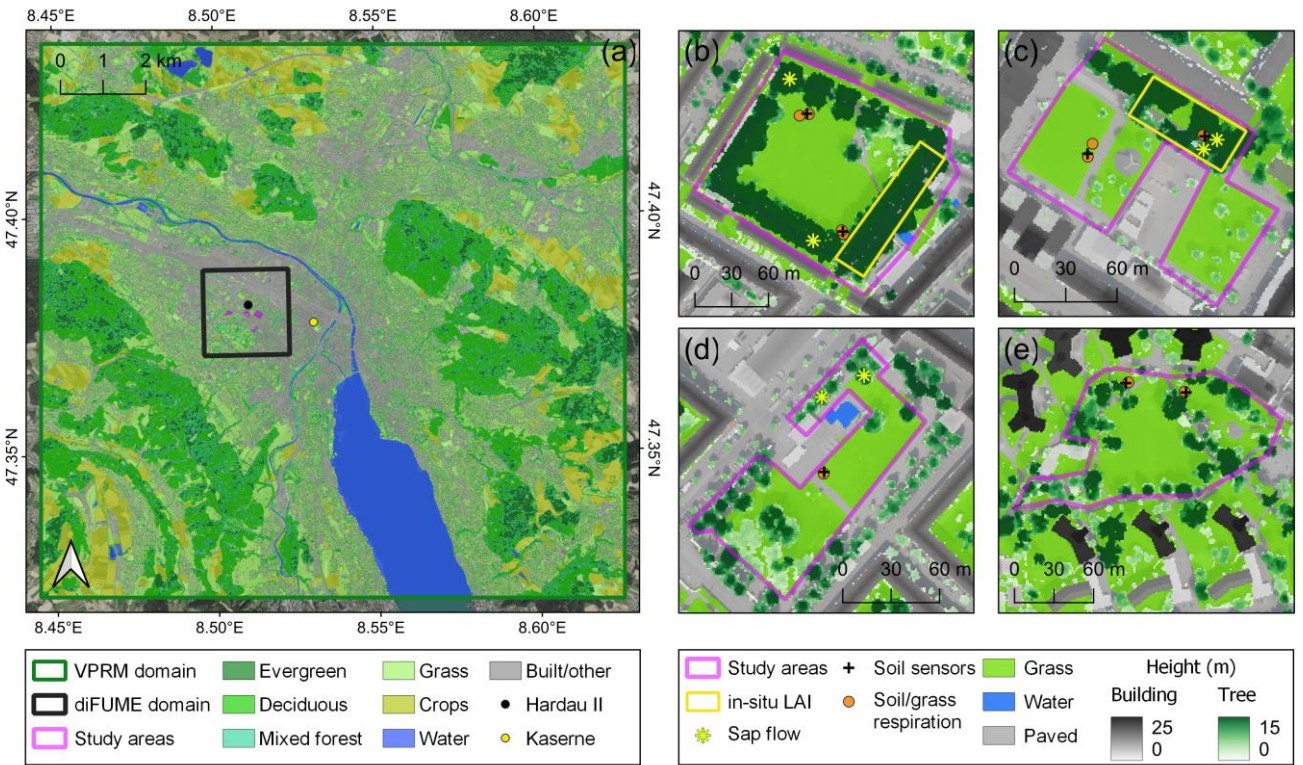

**Figure 1: (a) Locations of the study areas, meteorological stations (Hardau II, Kaserne) and the model domains over the land cover map of the Zurich region. Locations of the in-situ observations in (b) Bullingerhof, (c) Hardaupark, (d) Fritschiwiese, and (e) Heiligfeld parks over the land cover map and with the building and tree heights overlaid. All maps are projected in the Swiss CH1903+/LV95 coordinate system (EPSG: 2056) in a semi-transparent format over a basemap of aerial orthophotos (Orthofoto, 2024).**

## 2 Methods

### 2.1 Site description

The in-situ observations were performed in four public parks in the city district Hard in Zurich, Switzerland, close to the Hardau II ICOS-Cities tower site (Fig. 1). According to the climate normal of the period 1991 - 2020 from the weather station Fluntern (556 m a.s.l., 47.38° N / 8.57° E), the annual total precipitation in Zurich is 1108 mm, with summer months being rainier than winter months, and the annual mean temperature is 9.8 °C, reaching 24.3 °C monthly mean of daily maximum in July and -1.4 °C monthly mean of daily minimum in January. The four parks are similar in size, land cover types, use and management practices but they differ in terms of age, tree species, tree size/age and tree cover fractions (Table 1). Bullingerhof was developed in the 1930's and consists of rows of *Platanus* sp. trees, surrounding a large open lawn (Fig. 1b). Hardaupark is a more recently developed park. It features an old tree row of *Platanus* sp. in the northern part, which were planted in the 1970's, but the lawns which are the main parts of the park are covered only by small trees (*Larix x marschlinsii*, *Robinia pseudoacacia*), established since 2010 (Fig. 1c). Part of the lawn was previously a parking lot. Fritschiwiese was initially developed as a park in 1921, previously being part of the nearby cemetery. The northern part was redesigned in the 1970s due to underground constructions, which include energy generation units. Trees featuring mainly *Tilia* sp. are scattered and mainly surrounding the lawn in the centre, which were planted between the 1950s and 1970s (Fig. 1d). Heiligfeld was developed in 1955 and features a diverse array of tree species and ages, with *Carpinus betulus*, *Betula pendula*, *Acer* sp., *Pinus* sp. and *Quercus* sp. constituting the main park's tree population. Similar to the other parks, the central area of the park consists of an open lawn (Fig. 1e). The four parks are neither irrigated nor fertilised throughout the year. Grass is mowed frequently during spring and summer period, especially in the central areas of the parks, which are used by residents for recreation and sports. Some parts of the parks are not mowed, such as the east part of Heiligfeld.

Table 1. Main characteristics of the parks used as study areas. The surface cover statistics are based on the land cover product (Section 2.4) and the tree information comes from the city tree inventory (Baumkataster Zurich, 2024).

|  | Bullingerhof | Hardaupark | Fritschiwiese | Heiligfeld |
|---|---|---|---|---|
| Surface cover fractions (%) |  |  |  |  |
| Trees | 54 | 29 | 38 | 41 |
| Grass | 39 | 53 | 51 | 52 |
| Paved | 7 | 18 | 11 | 7 |
| Main tree species | *Platanus* sp. | *Platanus* sp. | *Tilia* sp. | diverse |
| Mean tree height (m) | 29 | 14 | 16 | 20 |
| Park age (years) | 94 | 14 | 74 | 59 |

### 2.2 In-situ observations

#### 2.2.1 Trees

Tree ecophysiological dynamics in the studied parks were monitored by measuring sap flux density on representative trees and leaf area index (LAI) at dense tree stands. Six trees were equipped with heat pulse sap flow sensors (3 x 3 cm probes, Implexx Sense) since July 2022, providing continuous measurements at 10-min sampling intervals. The sensors, along with the boxes containing power and electronics, were installed on the tree trunks at 2.9 – 3.5 m height a.g.l. to avoid vandalism. One sensor per tree was installed, always facing south and placed beneath the tree branches (except the northern tree in

Bullingerhof where the sensor is above the lowest branch). Two trees per park – except Heiligfeld – were equipped with sap flow sensors, two *Platanus* sp. trees in Bullingerhof, two *Platanus* sp. trees in Hardaupark and two *Tilia* sp. trees in Fritschiwiese (Fig. 1b-d). For each tree, the trunk diameter and the bark depth at installation height were measured at the time of installation. Heat velocity (m s$^{-1}$) was estimated from each sensor using the outer and inner needle thermistors at 0.5 cm and 1.5 cm sapwood depth using the heat ratio method and converted to sap flux densities (cm$^3$ cm$^{-2}$ h$^{-1}$), as described in

Forster (2019) and Forster (2021). The sap flux densities from the outer thermistors were consistently lower than the inner thermistors. Since sap flux density was used as a proxy for tree gross photosynthesis in this study, only the inner thermistor data was used in the analyses. The 10-min data per tree were averaged to hourly values per park to extract further statistics in this study. Given that sap flux density is controlled by similar environmental parameters as gross photosynthesis, these observations were treated in this study as a proxy for GPP.

LAI was measured in dense tree stands, found only in Bullingerhof and Hardaupark (Fig. 1b,c). LAI was measured since April 2022 only during sunny conditions using a ceptometer (SS1 SunScan, Delta-T Devices). The measurement frequency depended on the sky conditions, optimally with weekly repetitions during leaf-on and leaf-off periods and biweekly during the rest of the growing period. The photosynthetically active radiation (PAR) above and below the tree canopy was sampled sequentially using the ceptometer probe. Above-canopy readings were sampled in the open lawns near the tree stands. The

above canopy incident total and diffuse PAR was sampled by casting a small shadow on part of the probe according to the instrument protocol (Webb et al., 2016). Between 15 and 20 below-canopy PAR readings were subsequently sampled every ~2 m standing in the centre of the stand minor axis and moving across the major axis holding the probe perpendicular to the direction of the sun. Above-canopy incident PAR was sampled again after the below-canopy readings to ensure that the light conditions have not changed during each measurement cycle. LAI of each stand was estimated according to the instrument

software (Webb et al., 2016), assuming spherical leaf angle distribution and leaf PAR absorption of 0.85.

#### 2.2.2 Lawns

The in-situ measurements performed to monitor the park lawn dynamics included the installation of soil temperature (T$_{soil}$) and water content (SWC) sensors and the repeated measurement of soil and grass respiration. Seven soil sensors (TEROS 12, METER Group) were installed during May 2022 at different locations across the parks with the central needle (temperature

sensor) positioned at ~15 cm depth. The sensors' sensitivity to soil water content is contained within a ~1 L of soil volume around the sensor and were installed along a vertical axis to increase the measurement volume along this axis. According to the installation characteristics and the sensor specifications, the measurement volume of the sensors was between 9 and 21 cm depth. The sensors were buried in the soil along with the electronics and communication boxes. This allowed the selection of the sensor locations without any restrictions. The communication was achieved through LoRaWAN sensor

devices (Decentlab GmbH) with data transmission every 10 min.

Soil and grass respiration were measured near each soil sensor since July 2022 using a portable $CO_2$ soil efflux system equipped with a 20 cm diameter survey chamber (LI-8200-01S, LI-COR Biosciences) and a $CO_2$/$H_2O$ analyser (LI-870, LI-COR Biosciences). Ten soil collars were permanently installed at the four parks (Fig. 1), allowing repeated measurements at the exact same location under minimum soil disturbance. The collars were inserted completely into the ground to avoid any

interference with the park visitors and the maintenance activities (e.g. grass mowing). A removable 20 cm diameter metal adapter of 3.5 cm height was used as a fixing between the top part of the soil collar and the survey chamber. Some of the collars were intentionally left undisturbed to measure total grass and soil respiration ($R_{eco}$). Conversely, the grass in the rest of the collars was removed to enable the isolated measurement of bare soil respiration ($R_{soil}$). Only $R_{eco}$ was measured during the first two campaigns in summer 2022 and then gradually grass was removed from eight collars to measure $R_{soil}$. Two $R_{eco}$

collars were kept throughout winter (1x Hardaupark, 1x Heiligfeld) and two more (2x Bullingerhof) were established in June 2023. The aim of the different treatment strategies was to assess both, bare soil respiration and lawn $R_{eco}$ under similar environmental conditions. The $CO_2$ flux was calculated from the chamber measurements using an exponential fit to the increase of dry $CO_2$ concentration over time while the chamber was closed using SoilFluxpro software (LI-COR Biosciences). The derivation of the $CO_2$ flux equation is described in detail in the LI-8200-01S manual (LI-COR, 2024). The

total measurement time (chamber closed) was 2 min and a deadband of 25 s was applied. At least two repeated observations were conducted at each collar to account for potentially problematic measurements. $T_{soil}$ and SWC at 5 cm depth were measured at the same time with the chamber measurements next to each collar with the survey chamber integrated soil probe (HydraProbe, Stevens) additionally to the permanent soil sensors at 15 cm, to get a more complete overview of the soil conditions. Measurements of $R_{eco}$ and $R_{soil}$ were performed during day campaigns (07:00 - 16:00 CET) every two weeks. The

lawn measurement locations were selected to account for the variability of the environmental conditions and eventually of the soil and grass respiration fluxes across the parks. Sunny locations were selected in Bullingerhof, Hardaupark and Fritschiwiese. Additionally, shaded or partly shaded locations were selected in Bullingerhof, Hardaupark and Heiligfeld (Fig. 1).

## 2.3 Model description

Four different types of biosphere models were selected for this study; a full carbon cycling ecosystem model designed for natural ecosystems (JSBACH), a land surface model designed for urban areas (SUEWS), a satellite-based semi-empirical $CO_2$ flux model designed for urban areas (diFUME) and a satellite-based light-use-efficiency model initially designed for

natural ecosystems (VPRM). The rationale behind the model selection was to investigate if their performance in simulating the urban $CO_2$ exchanges follows the level of sophistication (i.e. the model complexity and detail in simulating the processes that govern $CO_2$ exchanges) and at the same time explore the advantages of the models that are specifically designed for urban applications over the ones designed for natural ecosystems. Furthermore, the selection of these models allows the comparison of different model attributes (e.g. satellite-based phenology versus modelled phenology), but also the identification of the challenges or drawbacks when applying models designed for natural ecosystems on urban environments. The models are described below in alphabetical order and a more detailed overview of each model can be found in the Appendix A.

### 2.3.1 diFUME

diFUME is a recently developed urban $CO_2$ flux model (Stagakis et al., 2023a), which uses a semi-empirical approach for modelling the biogenic $CO_2$ fluxes in urban areas based on the main environmental controlling parameters of photosynthesis and respiration processes (i.e. incoming global radiation, air-soil temperature, vapour pressure deficit, soil water content), capturing the spatiotemporal dynamics of plant phenology using high resolution satellite imagery and simulating the effects of urban morphology on the canopy radiation interception based on a voxel traversal (ray tracing) algorithm (Fig. A1). The study area is represented in three dimensions as a 3D grid of voxels of certain size (5 m in this application) and category (i.e. terrain, building, vegetation, air) according to the digital surface model (DSM) and land cover information. GPP is modelled for the vegetation voxels of each horizontal layer ($i$) based on the PAR reaching the sunlit and the shaded fraction of LAI in each voxel using a nonrectangular hyperbolic function and other empirical functions for the simulation of air temperature, vapour pressure deficit and soil water content effects. $R_{eco}$ is separated into the aboveground and belowground components; the aboveground is modelled based on an exponential fixed-$Q_{10}$ equation using air temperature, multiplied by LAI, and the belowground based on a modified Arrhenius equation using soil temperature and soil water content. The main equations used by diFUME model, as well as a model flow diagram are presented in Appendix A. The model version applied in this study contains some modifications compared to the version described in Stagakis et al. (2023a), such as the addition of a sky-view factor (SVF) estimation module using the voxel traversal algorithm, the addition of a module to split global radiation into its diffuse and direct components, and the simplification of the equations for the estimation of the diffuse and reflected PAR reaching the leaf surface of each vegetation voxel (i.e. directional SVFs and SVFs including tree canopies are not used in this version).

The model inputs used in this study are the land cover map, meteorology and DSMs, as described in Section 2.4. The LAI dynamics comes from the Copernicus High Resolution Vegetation Phenology and Productivity (HR-VPP) product, which includes four daily vegetation indices (PPI, NDVI, LAI and FAPAR) and quality information at 10 m resolution. The Copernicus LAI product was filtered per pixel according to the quality flags for clouds, cloud shadows, proximity to clouds and cloud shadows and snow, resampled to 5 m according to the land cover information (Stagakis et al., 2023a) and a 16-day maximum value composite filter was applied to exclude bad values and gaps. The 16-day product was then linearly

interpolated to derive 5-day LAI values which are used as model inputs. The maximum acceptable LAI value over grass pixels was set to 3 m$^2$ m$^{-2}$ since the park grass is frequently mowed.

### 2.3.2 JSBACH

JSBACH (Reick et al., 2013) is the land component in the Earth system models of the Max-Planck Institute for Meteorology
that simulates terrestrial energy, hydrology and carbon fluxes utilizing a number of submodels (Fig. A2). The vegetation is described by plant functional types (PFT). In this study, the vegetation was described by the C3 grass and broadleaved deciduous trees PFTs. The carbon assimilation is described by the biochemical photosynthesis model of Farquhar et al. (1980). The assimilation rate is limited either by the carboxylation rate ($J_{C,stress}$) or the electron transport rate ($J_{E,stress}$) (Table A1). $J_{C,stress}$ is a function of the maximum carboxylation rate ($V_{max}$), which has an Arrhenius-type temperature
dependence and $J_{E,stress}$ is a function of PAR (non-rectangular hyperbolic dependence) and $J_{max}$ (maximum electron transport rate), which has a linear dependence on temperature. Both $V_{max}$ and $J_{max}$ are inhibited above 55 °C. $V_{max}$ is also scaled with a factor depending on the canopy depth to account for the Rubisco profile in the canopy. Dark respiration ($r_d$) is a fixed fraction of $V_{max}$ at 25 °C with an Arrhenius-type temperature dependence. It is inhibited above 55 °C and decreased with increasing solar irradiance. The unstressed stomatal conductance ($g_L^{H_2O}$) is scaled by the plant available soil water ($f_{ws}$)
to obtain stomatal conductance under water stress ($g_{L,stress}^{H_2O}$), which is used to derive $J_{C,stress}$ and $J_{E,stress}$ and then finally the assimilation rate under water stress ($A_{stress}$). $f_{ws}$ depends on soil moisture in the root zone and specific humidity. The seasonal development of LAI is described by the Logistic Growth Phenology (LoGro-P) model (Böttcher et al., 2016). The development of LAI of broadleaved deciduous trees is described by the summer green phenology (Table A1). It has three phases, the growth period in the spring ($k > 0$ and $p = 0$), the vegetative phase during the summer ($k = 0$ and small $p$) and
the rest phase starting in autumn ($k = 0$ and high $p$). The transition from the rest phase to the growth phase is dependent on the evolution of the temperature using the alternating model of Murray et al. (1989), while the growth phase has a fixed duration. The maximum LAI is given as a parameter. The grass phenology further includes soil moisture and net primary productivity as determining factors. In the autumn, the phase transition occurs when the pseudo soil temperature (running mean of air temperature) falls below a critical soil temperature, while grasses grow when there is sufficient soil moisture and
temperature. Leaves are shed when NPP is negative. The soil hydrology parameters are set on the basis of soil texture. The soil moisture is simulated with five layers within a multilayer soil hydrological scheme. The dynamics of litter and soil carbon is described by the submodel Yasso07 (Tuomi et al., 2009, 2011). Five carbon pools are distinguished based on their chemical properties (acid hydrolyzable, water soluble, ethanol soluble, neither hydrolyzable nor soluble, humus). The first four pools (a,w,e,n) are tracked both above and below ground, and separate pools are used for woody and non-woody litter,
altogether 18 pools. The litter pools receive carbon input from vegetation through the litter flux, faeces from grazing and losses from reserve pool (Table A1). Decomposition of the litter pools causes carbon to transfer between the pools and to the atmosphere. The loss rates depend on temperature, water availability and type of litter elements. The JSBACH model is

forced by air temperature, total precipitation, shortwave and longwave radiation, air humidity, wind speed and $CO_2$ concentration, as described in Section 2.4.3.

### 2.3.3 SUEWS

The Surface Urban Energy and Water Balance Scheme (SUEWS, Järvi et al., 2011, 2019) is an urban land surface model simulating jointly the surface energy and water balances, and carbon dioxide surface exchange at the local or neighbourhood scale (Fig. A3). SUEWS encompasses various submodels, which account for factors like net all-wave radiation, heat storage (Grimmond et al., 1991; Sun et al., 2017), anthropogenic heat flux, irrigation, roughness layer temperature and relative humidity (Tang et al., 2021). These submodels are essential for accurately representing urban characteristics affecting the simulated balances and exchanges. Photosynthetic uptake is calculated using an empirical canopy-level photosynthesis model, where the potential photosynthesis is modified for different environmental factors (Table A1). The same environmental factors are also controlling the surface (stomatal) conductance. The seasonal development of LAI depends on the growing degree days (GDD) and senescence degree days (SDD), which depend on air temperature. The rates of leaf-on and leaf-off and maximum LAI are given as an input. During the leaf-on period, LAI remains relatively stable and responds slowly to stress conditions. In contrast, other environmental factors that regulate surface conductance exhibit more pronounced responses to stress. Soil and vegetation respiration is calculated as simple exponential dependence on air temperature. SUEWS provides a holistic approach on joint energy, water, and $CO_2$ cycles in urban environments, allowing to account, for example, for the influence of elevated air temperatures on water and $CO_2$ cycles. The model relies on standard meteorological inputs like wind speed, air temperature, air pressure, precipitation, and short-wave radiation. The forcing air temperature needs to be from above roughness sublayer, so in this study the measured 2 m air temperature is scaled to 35 m using lapse rate of 6.5°C/km. SUEWS is able to calculate 2 m temperature from the forcing temperature (Tang et al., 2021), which is used in the calculations of photosynthesis and soil and vegetation respiration. Additionally, site-specific data, such as surface cover fractions, tree and building heights, are needed. SUEWS has extensively been evaluated in simulating urban $CO_2$ exchanges in various cities including Helsinki, Minneapolis, Swindon and Beijing (Havu et al., 2022; Järvi et al., 2019; Zheng et al., 2023). For this study, we employed the latest available version SUEWS V2020a, using the land cover, meteorology and surface morphology inputs described in Section 2.4.

### 2.3.4 VPRM

The Vegetation Photosynthesis and Respiration Model (VPRM) is a satellite-based, data-driven model to estimate spatial and temporal surface biogenic $CO_2$ fluxes for different plant functional types (Mahadevan et al., 2008) (Fig. A4). The VPRM model represents the NEE of $CO_2$ as a combination of two components: GPP and $R_{eco}$. GPP is a light-dependent term using remote sensing vegetation indices, including the Enhanced Vegetation Index (EVI) and Land Surface Water Index (LSWI), combined with shortwave solar radiation to estimate the carbon uptake from photosynthesis. $R_{eco}$ is a light-independent part using only temperature at 2 m above the ground and a linear model to denote the carbon emission from the ecosystem

respiration (Table A1). Additionally, the VPRM requires an accurate vegetation cover map to provide the spatial distribution of different vegetation types, and utilizes distinct parameters for different vegetation types to drive the VPRM. These parameters are fitted by using the NEE observations from eddy covariance towers that monitor specific vegetation types. The VPRM model has been widely applied across Europe to estimate the large-scale biogenic $CO_2$ fluxes (Ahmadov et al., 2009; Gerbig and Koch, 2023; Zhao et al., 2023).

For the vegetation indices, we used the Sentinel-2 MSI - Level 2A (MAJA Tiles) product (10 m spatial resolution) provided by the German Aerospace Center (DLR, 2019). The vegetation cover is taken from our self-developed vegetation land cover product, which is described in Section 2.4.1, and has been resampled to align with the Sentinel-2 image grid. Temperature and shortwave downward solar radiation data are sourced from the Zurich Kaserne station (Section 2.4.3). The plant type specific parameters for VPRM are those presented by Zhao et al. (2023).

## 2.4 Data inputs

### 2.4.1 Land cover

For the accurate modelling of high spatial resolution biogenic fluxes, a precise and high-resolution vegetation land cover map is needed for diFUME, SUEWS and VPRM models (Table 2). The following high-quality datasets available in Zurich made this objective possible:

a. Land Use Cadastre of the Canton of Zurich (Amtliche Vermessung, 2024): This detailed GIS dataset is the official land registry of the Canton and the basis for geographical information in almost all domains from spatial planning to agriculture and tourism. It acted as our base map, with vegetation types reclassified to distinguish between grassland and cropland areas. Note that the 'closed forest' was assigned to grassland since it was replaced by more accurate tree cover data later while also retaining the grassland information within the forests.

b. Urban Atlas (Urban Atlas, 2018): For deriving the cropland coverage, we utilized the Urban Atlas 2018 from the Copernicus Land Monitoring Service. This data set discriminates between crop and grass fields, which was not possible based on the Canton Cadastre alone.

c. Vegetation Height Model (VHM) from the Swiss federal forest inventory (Vegetationshöhenmodell LFI, 2019): This dataset is based on a combination of stereo aerial images and Lidar observations and is available for the whole country at a resolution of 1 m x 1 m. Pixels indicating trees shorter than 2 meters were excluded from this dataset, as these pixels, based on our observations, often are noisy signals.

d. Forest Mixture from the Swiss Federal Forest Inventory (Waldmischungsgrad LFI, 2018): This dataset derives the forest mixture ratio based on Sentinel-1 and Sentinel-2 satellite observations and is available for the whole country at a resolution of 10 m x 10 m. Two empirical thresholds were used to convert the forest mixture ratio to three tree types: deciduous (greater than 80%), mixed forest (between 20% and 80%), and evergreen (less than 20%). Within the city of Zurich, a large majority of trees is deciduous.

We merged tree classifications with the VHM to produce a detailed tree species cover map. Areas that were unidentifiable, often found in urban regions, were labelled as deciduous. The generated tree species cover map was then combined with the other datasets, resulting in our 1-meter resolution land cover map for Zurich.

**2.4.2 Digital Surface Models**

The very high resolution DSM products (terrain, building and tree heights) developed by the Zurich city authorities (Baumhöhen, 2023; Digitales Oberflächenmodell, 2022; Digitales Terrainmodell, 2022) were used in this study as inputs for the diFUME and SUEWS models (Table 2). The DSM products are derived from high-resolution laser scanning (LIDAR, average point density of 16 points/m$^2$) over the years 2021 and 2022.

**Table 2: Overview of data inputs required by each model during the simulation period 01/2022–09/2023.**

| Input | Time-step, aggregation | Location | diFUME | JSBACH | SUEWS | VPRM |
|---|---|---|---|---|---|---|
| Air temperature ($T_{air}$), 2 m | Hourly, average | Kaserne | X | X | X | X |
| Global radiation | Hourly, average | Kaserne | X | X | X | X |
| Downward longwave radiation | Hourly, average | Hardau, ERA-5 | | X | | |
| Relative humidity, 2 m | Hourly, average | Kaserne | X | X | X | |
| Wind speed, 35 m | Hourly, average | Kaserne | | X | X | |
| Air pressure, 2 m | Hourly, average | Kaserne | | | X | |
| Precipitation | Hourly, total | Kaserne | | X | X | |
| Atmospheric $CO_2$ concentration | Monthly, average | Hardau, Beromunster, ERA-5 | | X | | |
| Soil temperature ($T_{soil}$) | Hourly, average | Parks | X | | | |
| Soil water content (SWC) | Hourly, average | Parks | X | | | |
| Land cover map | Static | City | X | | X | X |
| DSM | Static | City | X | | X | |
| Satellite LAI | Daily, linear interpolation | City | X | | | |
| Satellite EVI, LSWI | Daily, linear interpolation, Savitzky-Golay filter | City | | | | X |

### 2.4.3 Weather driver data

We used meteorological measurements mainly from the site Zurich Kaserne (01/2022–09/2023), which is a station of the Swiss national air pollution monitoring network NABEL. The measurements are performed with the same high-quality instruments as used in the national weather observation network Swiss MetNet. Zurich Kaserne is located in a large courtyard at a distance of about 1 - 1.5 km from the parks analysed in this study (Fig. 1). The measurements from Zurich Kaserne are expected to be representative for the parks, as it is located in a similar urban setting. Wind and global radiation

are measured on top of a neighbouring four-storey building. Wind is measured at 35 m and global radiation at 27 m above ground. Additionally, downward longwave radiation and atmospheric $CO_2$ concentration data was derived for the period 07/2022–09/2023 from the ICOS-Cities Hardau II station (110 m a.g.l.). For the period prior to the Hardau II installation (01/2022–06/2022), as well as for the spin-up period of the JSBACH model (1950–2021), the Copernicus ERA5-Land dataset was used to derive all missing observations with the exception of $CO_2$ concentration, which was derived from

Beromunster station (11/2012–02/2022) and prior to that from a global gridded monthly data set of $CO_2$ (Cheng et al., 2022). The $CO_2$ data was used in JSBACH as monthly means. The meteorological measurements used as input for the models during the simulation period 01/2022–09/2023 are listed in Table 2. The diFUME model additionally requires $T_{soil}$ and SWC, which were measured in seven locations across the studied parks as described in Section 2.2.2.

### 2.5 Model comparison and evaluation

All models were run for the period 01/2022–09/2023 for the four parks using the hourly meteorological inputs as described in Table 2. diFUME and VPRM ran spatially at high resolution (5 m and 10 m respectively), while SUEWS and JSBACH provided integrated simulations for each land cover type of each park. This intercomparison study focuses on the park trees and lawns separately. To compare between the spatial and integrated model simulations, the grass and tree pixels of each park from diFUME and VPRM outputs were selected according to the land cover map. All pixels covered by at least 90 %

by a given land cover type based on the 1m land cover map were selected and averaged for each park. Hourly GPP, $R_{eco}$ and NEE estimates for each park were then temporally averaged to daily, monthly or annual values. $CO_2$ flux units were kept uniform across the manuscript ($\mu$mol m$^{-2}$ s$^{-1}$) except for the annual totals where the units were converted to kg m$^{-2}$ a$^{-1}$ to facilitate the comparison with relevant literature. In this study, we kept GPP and $R_{eco}$ positive for easier plotting and comparison and estimated NEE as NEE = $R_{eco}$ – GPP.

Given the available in-situ observations, which do not cover all carbon cycle components and drivers due to technical and logistical restrictions, we focused on the evaluation of tree LAI and GPP, as well as SWC and $R_{eco}$ in the lawns. Modelled LAI can be directly evaluated using the field observations in the measured tree stands. However, VPRM does not include LAI in the light-use-efficiency equation, but uses EVI as a proxy of vegetation greenness. Therefore, and in order to achieve uniform evaluation across the four models, we converted EVI to LAI using a linear regression ($R^2$ = 0.86, slope = 7.03,

intercept = -1.26) between the HR-VPP LAI and Sentinel-2 EVI for all cloud-free acquisitions within the study period using

the mean values over trees of each park. A similar approach was adopted for the evaluation of tree GPP using sap flow as a proxy. Since the diurnal behavior of sap flow is expected to be quite different from GPP, the evaluation was performed using daily integrated values. The daily averaged sap flow values ($cm^3$ $cm^{-2}$ $h^{-1}$) were converted to GPP ($\mu mol$ $m^{-2}$ $s^{-1}$) by a linear regression ($R^2 = 0.81$, slope = 0.77, intercept = 1.93) between the mean GPP by the four models over the four parks and the mean sap flow of all sampled trees.

The SWC observations and model estimates cannot be directly compared between them because of different soil layers considered and different soil parameterisations. Specifically, the observations were representative of approximately 9 to 21 cm soil depth, which can be closely compared to one soil layer simulated by JSBACH, but cannot be directly compared with SUEWS because the latter estimates a bulk SWC value for the total simulated soil volume. Furthermore, different soil parametrisations in the models can lead to different absolute maximum and minimum SWC values, which makes the direct comparison difficult (Fig. B1c). Therefore, the daily means of SWC observations and model estimates were normalised according to the individual time-series maximum and minimum values to avoid the effects of different model parameterisations for soil, as well as the effects of different soil depths measured and modelled. By applying this normalisation on SWC time-series, we focused on the evaluation of the temporal variability of each model rather than the absolute values. For VPRM, LSWI was used in this analysis as a proxy of SWC. Finally, lawn $R_{eco}$ was evaluated using the averages of the in-situ measurements for each daily campaign and the averages of the model outputs for the same days considering only the hours during the times of the measurements.

For the parameter evaluations, the average simulated value of the four parks (tree or lawn according to the parameter) of each model was compared to the average of the available observations, except for LAI, for which only simulations at Bullingerhof and Hardaupark were used was well as for $R_{eco}$ for which the observations at sunny and shaded locations (see Section 2.2.2) were averaged separately and the average of the two was considered as the representative $R_{eco}$ value to be compared with the models. The quantitative evaluation analysis in this study is presented using Taylor diagrams, combining three performance metrics: the normalized standard deviation, the correlation coefficient and the centred root-mean-square error (Taylor, 2001).

## 2.6 Uncertainty analysis

As a simple measure of model uncertainty of the different biogenic $CO_2$ flux components (GPP, $R_{eco}$, NEE), we computed the standard deviation of each hourly value of the four models. The hourly time-series of standard deviations were further aggregated to seasonal and diurnal patterns to examine the magnitude and significance of the biogenic model uncertainties. In order to address the importance of the biogenic $CO_2$ fluxes and their uncertainties in relation to the magnitude of the anthropogenic emissions, we focused on a central part of the city of Zurich corresponding to the diFUME model domain (Fig. 1) and estimated the monthly anthropogenic emissions, considering building heating, industry, vehicle traffic and human respiration. The monthly biogenic $CO_2$ fluxes (GPP, $R_{eco}$, NEE, averages of the four models) were scaled to the 2 km x 2 km domain according to the tree and grass land cover fractions and assuming that the tree and lawn simulations of the

four parks are representative of the vegetation of the domain. The anthropogenic emissions for the specific area were
estimated from the gridded annual inventory of the city of Zurich (Emiproc, 2024; Emissionskataster, 2024). For both
anthropogenic and biogenic fluxes, the monthly average of the two years of the study was estimated.

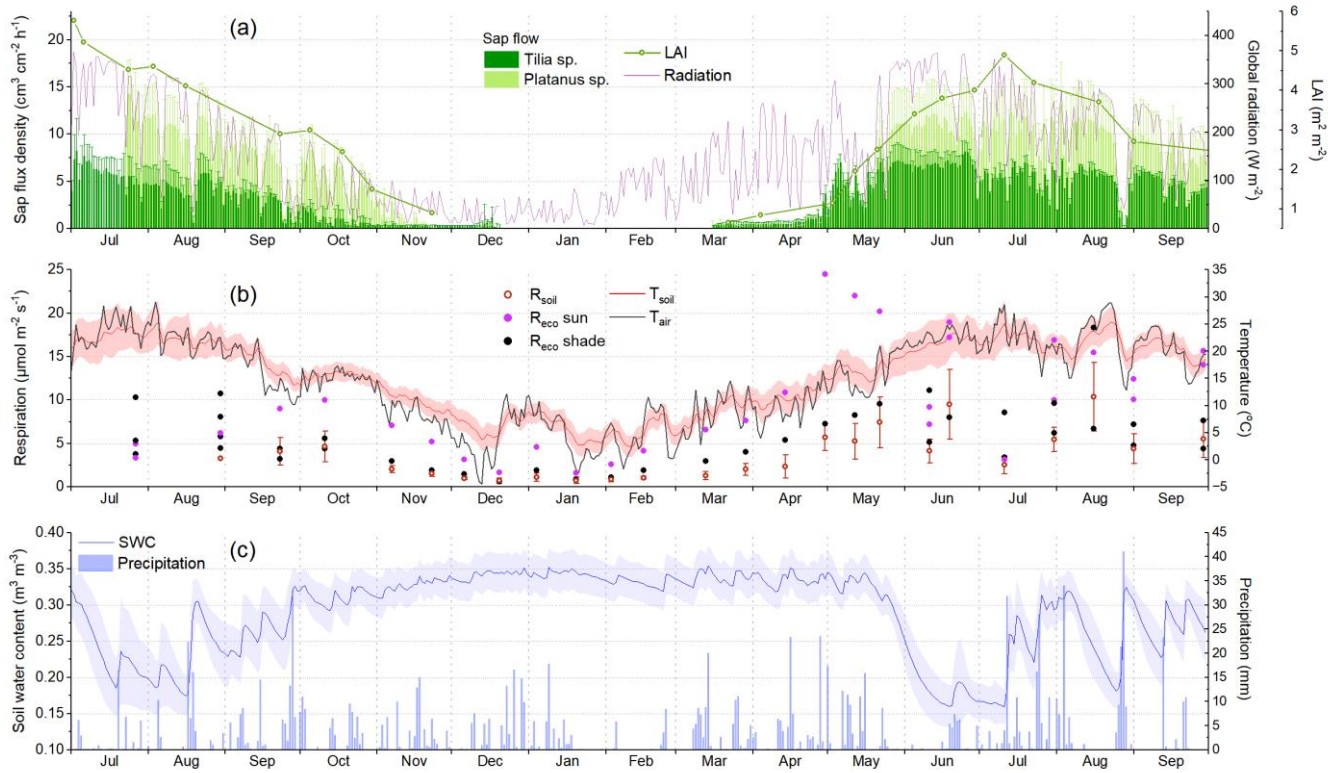

**Figure 2. Time-series of the in-situ observations in the study parks (Fig. 1) during the campaigns from May 2022 to Sep 2023. (a) Average leaf area index (LAI) from the two *Platanus* sp. stands, daily averages and standard deviations of the sap flux densities per tree species and daily average incoming global radiation. (b) Average and standard deviation of the measured soil respiration**
**($R_{soil}$), measured lawn ecosystem respiration ($R_{eco}$) (sunny and shaded locations in different colour) and daily average air and soil temperatures where the shading refers to the standard deviation of the seven soil sensors. (c) Daily average (line) and standard deviation (shade) of the measured soil water content (SWC) and daily precipitation (bars).**

## 3. Results

### 3.1 In-situ observations

Tree LAI and sap flow observations provided detailed information on tree phenology and physiology (Fig. 2a). The phenology of the *Platanus* sp. stands was somewhat different in the two study years. In 2022, leaf growth started in mid-April and already by mid-May, LAI had reached nearly 4 m² m⁻² (Fig. B1a). The peak was in the end of June, reaching around 5.5 m² m⁻². Then, LAI gradually declined until the end of November when the leaf fall was completed (Fig. 2a, B1a).

In 2023, the leaf growth started around the same time as in 2022 but was considerably slower. By mid-May, LAI had reached only about 2.5 $m^2$ $m^{-2}$ and peaked at the beginning of July at a lower value than in 2022 (~4.5 $m^2$ $m^{-2}$). The different phenological patterns observed in the two studied years were also confirmed by the satellite indices (Fig. B1a).

Sap flux density observations of the *Platanus* sp. followed the observed LAI seasonality but were very variable according to the environmental conditions, covarying strongly with incoming radiation, especially during summer periods (Fig. 2a). Sap

flux density of the *Tilia* sp. showed similar seasonal variability but consistently lower values except during spring 2023, when *Tilia* sp. trees had apparently earlier leaf growth than the *Platanus* sp. trees. On the other hand, *Tilia* sp. trees dropped their leaves a bit earlier than *Platanus* sp. trees according to the sap flow observations during October and November 2022 (Fig 2a). The effects of the drought periods (Fig. 2c) on the sap flow are not clearly discernible in Fig. 2a, however there was an increase of sap flow visible after the rains at the middle of August 2022.

The measurements at the park lawns provided valuable insights into the seasonal dynamics of grass and soil respiration. $R_{soil}$ seasonal variability was consistently following $T_{soil}$ and SWC changes (Fig. 2b,c). Applying a modified Lloyd and Taylor (1994) equation fit to the $R_{soil}$ observations, their variability was explained to 64 - 72 % by the $T_{soil}$ and SWC observations. $R_{soil}$ was low during winter, rose in spring and reached high values during hot summer days if soil moisture was available (Fig. 2b). During summer, conditions were more variable depending on the park location and hour of measurement and

therefore the measured $R_{soil}$ varied significantly. The highest $R_{soil}$ was measured during June and August 2023, reaching values above 10 μmol $CO_2$ $m^{-2}$ $s^{-1}$. The high value in June, which occurred during a rain event, is bracketed by much lower values during the prolonged drought period in June/July 2023.

Lawn $R_{eco}$ followed roughly the same seasonal pattern as $R_{soil}$ but with consistently higher values (Fig. 2b). Moreover, the variability observed in $R_{eco}$ between the different collars was very intense. This was to be expected since the grass biomass,

which contributed to the measured flux, was variable according to the park location. The general tendency observed on-site was that the shaded or partly shaded locations tended to have less thick and dense grass cover than the sunny locations. This was confirmed in the measurements by comparing shaded $R_{eco}$ with $R_{soil}$. The differences were not very big in the majority of the cases, indicating that $R_{soil}$ was the main source at these locations. On the other hand, shaded locations did not dry out so quickly as the sunny locations during drought conditions, maintaining the required soil moisture for healthy aboveground

leaf biomass. Therefore, it was observed in the measurements that shaded $R_{eco}$ was sometimes higher than sunny $R_{eco}$ during summer (Fig. 2b). However, the grass in sunny locations tended to recover quickly after rain events that restored the SWC during summer and autumn. Another important parameter to consider when interpreting the $R_{eco}$ observations is the grass mowing. Mowing has been reported to induce high respiration rates (Allaire et al., 2008; Kaye et al., 2005), which is the most probable explanation of the very high $R_{eco}$ measured in the sunny lawn locations since the end of April 2023 and during

summer months (Fig. 2b). The highest $R_{eco}$ values were found in the sunny location in the Hardaupark, which is the most recently constructed park in this study. Fitting the modified Lloyd and Taylor (1994) equation to the $R_{eco}$ measurements showed that the observed variability can be explained only to 38 % by measured $T_{soil}$ and water content.

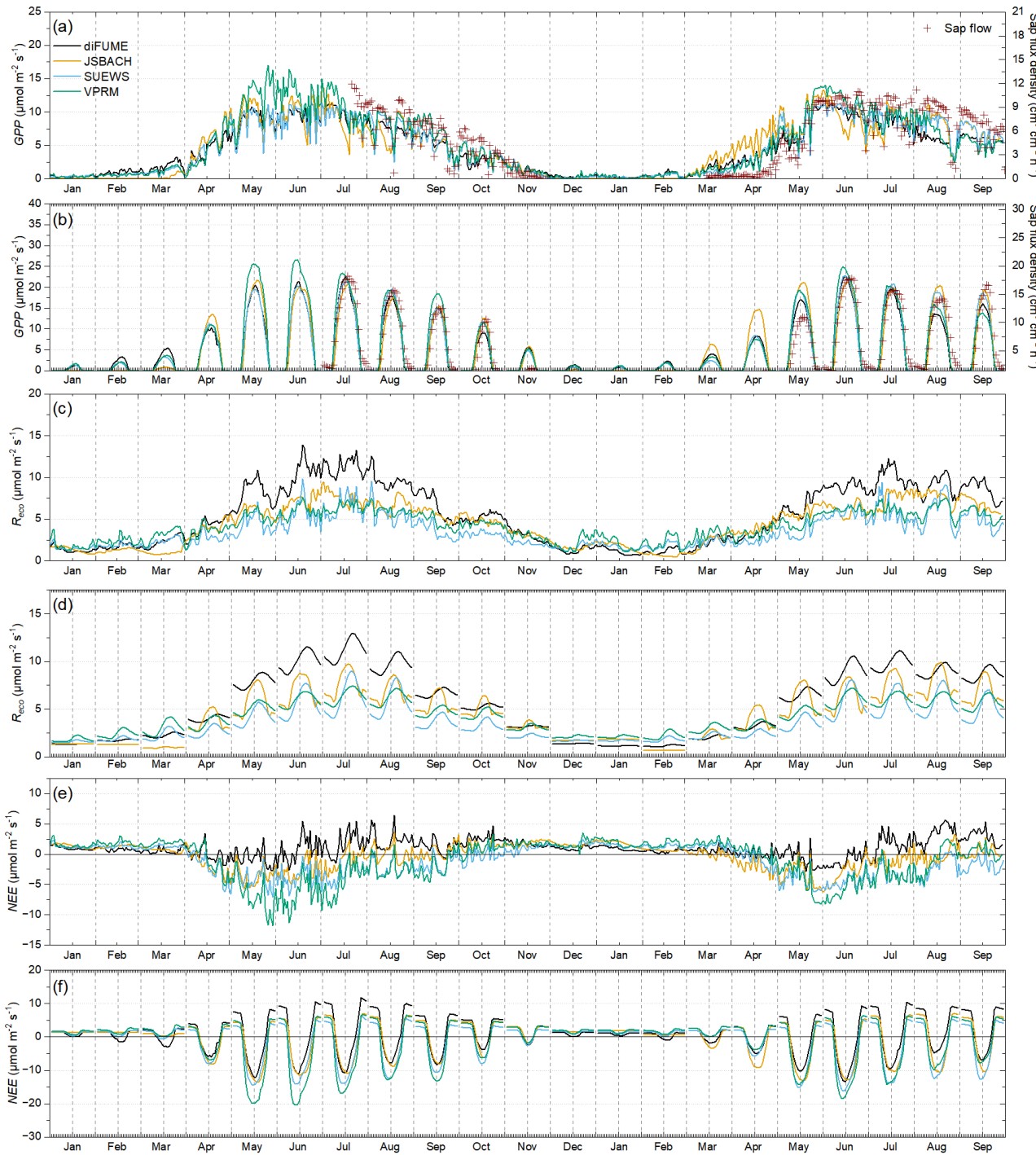

**Figure 3. Time-series of model outputs for park trees (averages of the four parks). Daily averages of (a) gross primary production (GPP), (c) ecosystem respiration (R_eco) and (e) net ecosystem exchange (NEE) estimates and diurnal hourly averages per month of**

**(b) GPP, (d) R$_{eco}$ and (f) NEE estimates. Sap flux density observations (averages of all trees) per day and hour are plotted over GPP in (a) and (b).**

## 3.2 Model comparison

The GPP, R$_{eco}$ and NEE outputs of the four models were compared for park trees (Fig. 3) and lawns (Fig. 4). Figures 3 and 4
500 contain time-series of daily totals and hourly diurnal averages of each month for each modelled parameter. Observed tree sap flow and lawn R$_{eco}$ are plotted against modelled tree GPP (Fig. 3a,b) and modelled lawn R$_{eco}$ (Fig. 4c), respectively. In general, models seemed to better agree for park trees than lawn. Tree GPP estimates were surprisingly consistent between the four models (Fig. 3a,b). Exceptions were the higher VPRM estimates between mid-May and mid-July 2022 and the higher JSBACH estimates during mid-March to mid-May 2023. A considerable deviation between the models was also
505 observed during August and September 2023, when diFUME and VPRM showed lower GPP than the other two models. Using sap flow as a GPP proxy, it was observed that the tree phenological and physiological seasonal dynamics were captured well by the models (Fig. 3a). The modelled diurnal GPP profiles agree also well with the sap flow observations, even though there were some differences in the timing (i.e. sap flow peaked later than GPP) as expected. However, there was considerable overestimation of spring GPP, especially during April and beginning of May 2023.

510 Lawn GPP estimations did not match well between the models (Fig. 4a,b). VPRM and JSBACH showed high lawn GPP during spring and early summer and strong variability from day to day according to daylight conditions. diFUME and SUEWS presented more conservative lawn GPP estimates, much lower than the other two models, and not so intense day to day variations. During mid-July – August 2022, all models captured the drop of lawn GPP due to drought conditions, with JSBACH showing reduced sensitivity of GPP to drought compared to the other models. During summer 2023, VPRM
515 showed a very intense drop of lawn GPP from the beginning of June due to drought conditions (see Fig. 2) and only a slight recovery after mid-July (Fig. 4a). Such intense variability was only captured by JSBACH, which simulated intense drops and fast recoveries after rain events during the whole summer of 2023. GPP seasonality in VPRM is mainly driven by the satellite EVI.

Simulated R$_{eco}$ seasonal variability agreed well between the four models (Fig. 3c, 4c). diFUME showed distinctively higher
520 R$_{eco}$ for trees during summer, while SUEWS showed lower R$_{eco}$ for lawns compared to the other models. diFUME showed very similar R$_{eco}$ estimated for lawns and trees, which was also the case for GPP, because in contrast to the other models, the parameterisation was the same for all plant types in the version used in this study. The rest of the models showed higher R$_{eco}$ for lawns during summer compared to trees. The diurnal patterns of R$_{eco}$ were slightly different between the models (Fig. 3d, 4d). JSBACH peaked earlier in the day and diFUME peaked later in the afternoon compared to the other models. Moreover,
525 diFUME tended to show higher R$_{eco}$ during night compared to the other models, most probably because T$_{soil}$ is the main driver of R$_{soil}$ in this model instead of T$_{air}$. None of the models seemed to capture any distinctive drought effect on lawn R$_{eco}$ (Fig. 4c). When compared to the lawn R$_{eco}$ observations, it appeared that the simulated values were closer to the shaded collar observations, while the sunny collar observations were much higher than the simulations during most of the time,

except during drought conditions when simulations were higher than the observations. It must be noted here that the comparison between observation and simulations shown in Fig. 4c is not entirely accurate because the observations were taken during a specific time of day and the daily averages take into account only the observation times. More detailed comparisons between $R_{eco}$ observations and simulations are presented in Fig. B2d. Overall, it can be deduced that all models were conservative in their lawn $R_{eco}$ estimations and were underestimating the full potential of $R_{eco}$ in open sunny park areas with thick managed lawn under wet and warm conditions.

NEE estimates of each model showed more distinctive seasonal and diurnal patterns compared to GPP and $R_{eco}$ alone (Fig. 3e,f, 4e,f). For both trees and lawns, VPRM showed the strongest (most negative) NEE values during spring and summer, except during late summer periods for lawns, when daily NEE turned abruptly to high positive values. diFUME was the most "conservative" model, with daily NEE values close to zero during the whole growing period for trees and lawns, with late summer NEE turning towards more positive values due to the drop of GPP and the consistently high $R_{eco}$. JSBACH and SUEWS daily NEE were intermediate between diFUME and VPRM estimates. In contrast to the rest of the models, SUEWS showed very different NEE estimates between trees and lawns with trees estimated as more productive than lawns. The diurnal NEE patterns were very much alike between the four models (Fig. 3f, 4f), but diFUME showed distinctively higher night-time NEE values during spring and summer, especially for tree areas, while SUEWS diurnal lawn NEE amplitude was particularly smaller than the other models.

The annual totals estimated from each model are presented in Fig. 5. There was good agreement in the total tree GPP but not for tree $R_{eco}$ where diFUME estimated the highest and SUEWS the lowest total $R_{eco}$ (Fig. 5a). As a result, diFUME estimated that park tree areas acted as sources of $CO_2$, while SUEWS estimated them to be sinks. JSBACH NEE was very close to zero with standard deviation that spanned between positive and negative values, while VPRM showed high GPP variability between parks and the standard deviation of NEE ranged also between positive and negative values. For the lawns, SUEWS showed distinctively lower total GPP and $R_{eco}$ compared to the other models. JSBACH and VPRM agreed well in the totals of GPP and $R_{eco}$, but VPRM showed much higher variability of total GPP. diFUME showed similar total lawn $R_{eco}$ with JSBACH and VPRM but lower total GPP. In total, all models agreed that park lawns acted as $CO_2$ sources. However, for VPRM the standard deviation was high and ranged between negative and positive values.

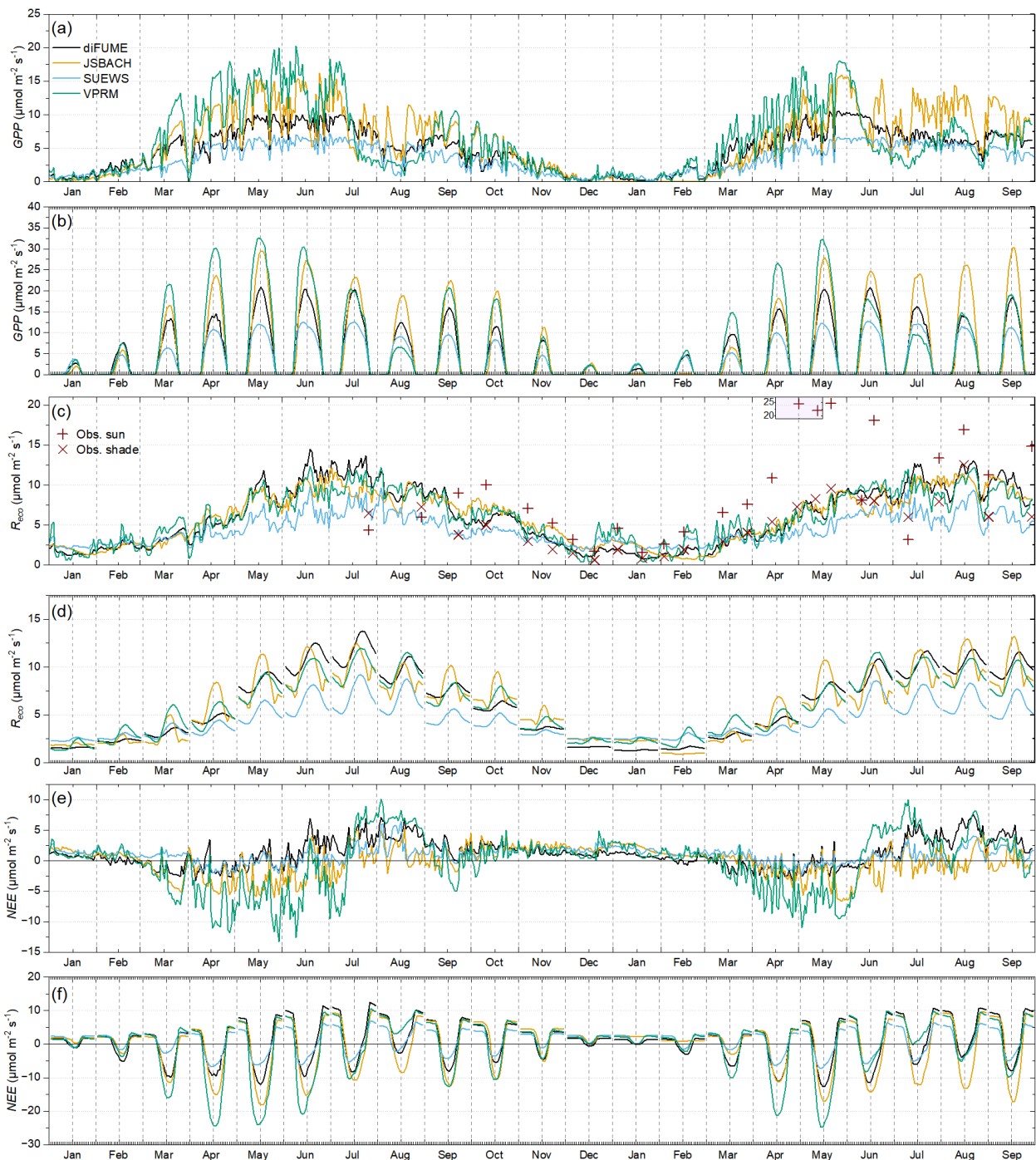

**Figure 4. Time-series of model outputs for park lawns (averages of the four parks). Daily averages of (a) gross primary production (GPP), (c) ecosystem respiration (R_eco) and (e) net ecosystem exchange (NEE) estimates and diurnal hourly averages per month of**

**(b) GPP, (d) R_eco and (f) NEE estimates. The observations of lawn R_eco (averages of sunny and shaded locations per campaign presented separately) are plotted over modelled R_eco in panel (c).**

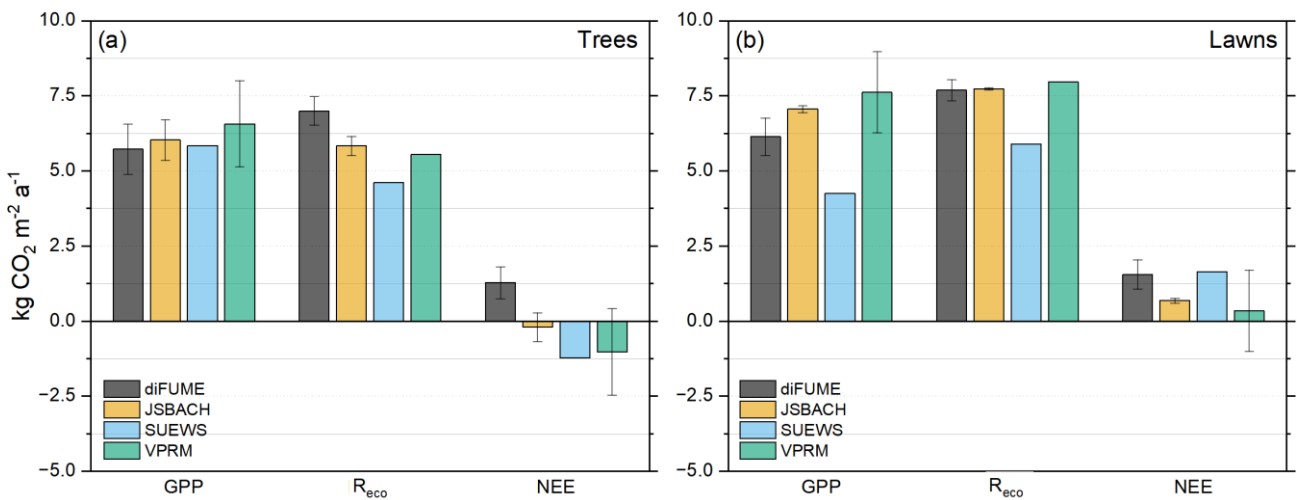

 **Figure 5. Annual totals for gross primary production (GPP), ecosystem respiration (R_eco) and net ecosystem exchange (NEE) for (a) trees and (b) lawns of the study parks. The whiskers express the standard deviation between the four parks. The period used to estimate the totals is between July 2022 and June 2023.**

### 3.3. Model evaluation

The Taylor diagrams provide a comprehensive intercomparison of the performance of the four models (Fig. 6). We focus only on certain parameters according to the available in-situ observations, as described in Section 2.5. It can be observed in Fig. 6a that all models underestimated tree LAI, but the two satellite-based models (diFUME and VPRM) captured the dynamics of tree LAI better than the process-based models (SUEWS, JSBACH), which showed lower correlations with the observations. More specifically, all models underestimated the high LAI values during mid-summer, while SUEWS consistently overestimated the tree LAI during the rest of the seasonal cycle (Fig. B1a, B2a). However, the performance with respect to LAI did not necessarily reflect on tree GPP simulation performance since all models captured well the measured daily sap flow variability (Fig. 6c, 3a), with the best performance demonstrated by SUEWS and the worst by JSBACH, most probably because of the early onset of GPP in JSBACH during spring 2023 (Fig. 3a). The lawn SWC variability (normalized) was captured very well by JSBACH and SUEWS (Fig. 6b), with the latter being a bit slower in the response to dryness and rain events (Fig B1c), most probably because SUEWS does not simulate multiple soil layers such as JSBACH. On the contrary, satellite-based SWC estimation by VPRM did not perform equally well. Even though LSWI roughly detected the dry periods during summer, it was not so accurate when it came to fast responses of SWC to rain events and revealed unrealistically low SWC during winter (Fig. B1c). All models underestimated lawn R_eco (Fig. 6d), especially SUEWS, while JSBACH showed better performance compared to the other models. The reason for the underestimation is mainly the high values of measured R_eco in the fully sunlit park areas during spring 2023 (Fig. 2b, 4c, B2d).

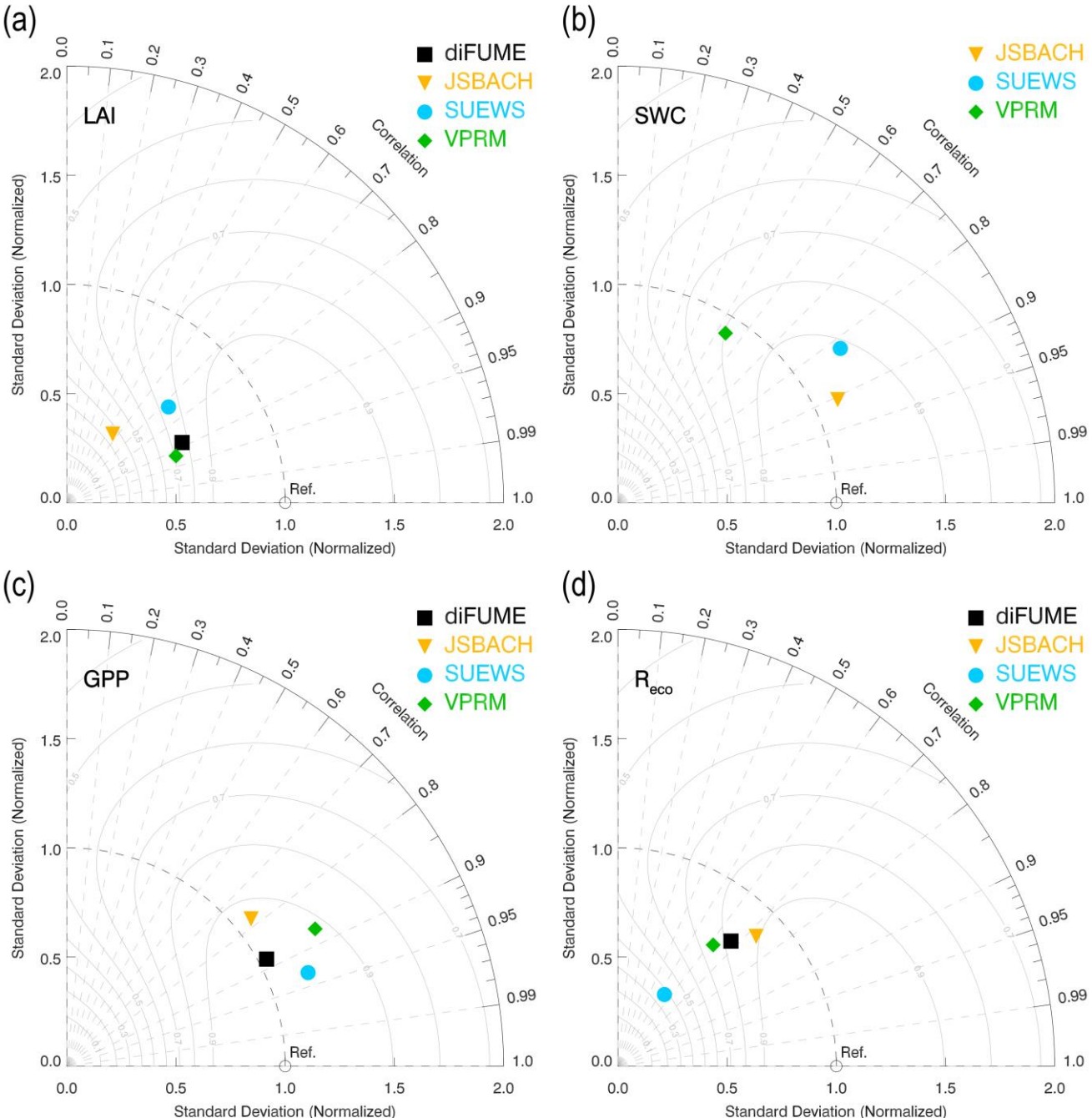

**Figure 6. Taylor diagrams describing the model performance in capturing the variabilities of (a) tree leaf area index (LAI), (b) lawn soil water content (SWC), (c) tree gross primary production (GPP), and (d) ecosystem respiration (R$_{eco}$) of lawns. The Taylor diagrams combine three performance metrics: the standard deviation (x, y axes, grey concentric cycles), the correlation coefficient (azimuth angle, grey lines) and the centred root mean-square error (dashed orange concentric cycles) (Taylor, 2001). The reference observation is plotted on the x-axis according to its standard deviation (red square) and the distance from this point is inversely related to the performance of each model (coloured dots).**

### 3.4. Biospheric flux uncertainty based on model spread

The dynamics of the estimated model uncertainties is presented in Fig. 7. Tree areas clearly showed lower uncertainties
compared to lawns mostly because of the increased inconsistencies between the models in the estimation of lawn GPP (Fig. 7b). The uncertainties for lawn GPP were highest during spring, reaching an average of around 9 $\mu$mol m$^{-2}$ s$^{-1}$ during midday. Tree GPP uncertainties were highest during summer, reaching around 4 $\mu$mol m$^{-2}$ s$^{-1}$ (Fig. 7a). Interestingly, the monthly diurnal patterns of GPP uncertainties of trees during the leaf-on period were different from lawns, presenting the highest values early in the morning or late in the afternoon rather than during midday. Early morning peaks in tree GPP
uncertainty appeared in June and July, while during the rest of the growing season the peaks appeared mostly during afternoon (Fig. 7a). This pattern is the result of differences in the diurnal evolution of GPP with some models having the maximum earlier in the day or showing wider peaks than others. $R_{eco}$ uncertainties were similar in magnitudes and patterns between trees and lawns (Fig. 7). They followed the seasonal growth of $R_{eco}$ with higher values during warm months. The diurnal variabilities of $R_{eco}$ uncertainties were not so pronounced in most of the cases, $R_{eco}$ uncertainty being close to zero
during winter and reaching maximum around 2.5 $\mu$mol m$^{-2}$ s$^{-1}$ during summer. It is interesting to note that during summer, $R_{eco}$ uncertainty tended to be highest during late evening hours and lower during the day, especially in trees. During the rest of the year, $R_{eco}$ uncertainties followed the normal $R_{eco}$ diurnal pattern with higher values during day and lower during night. As a result of the variable and high GPP uncertainties compared to $R_{eco}$, NEE uncertainties mainly followed the GPP uncertainty patterns during day and the $R_{eco}$ patterns during night (Fig. 7). This led to low NEE uncertainties during night
and high uncertainties during day with more pronounced diurnal variabilities for lawns and lower for trees.

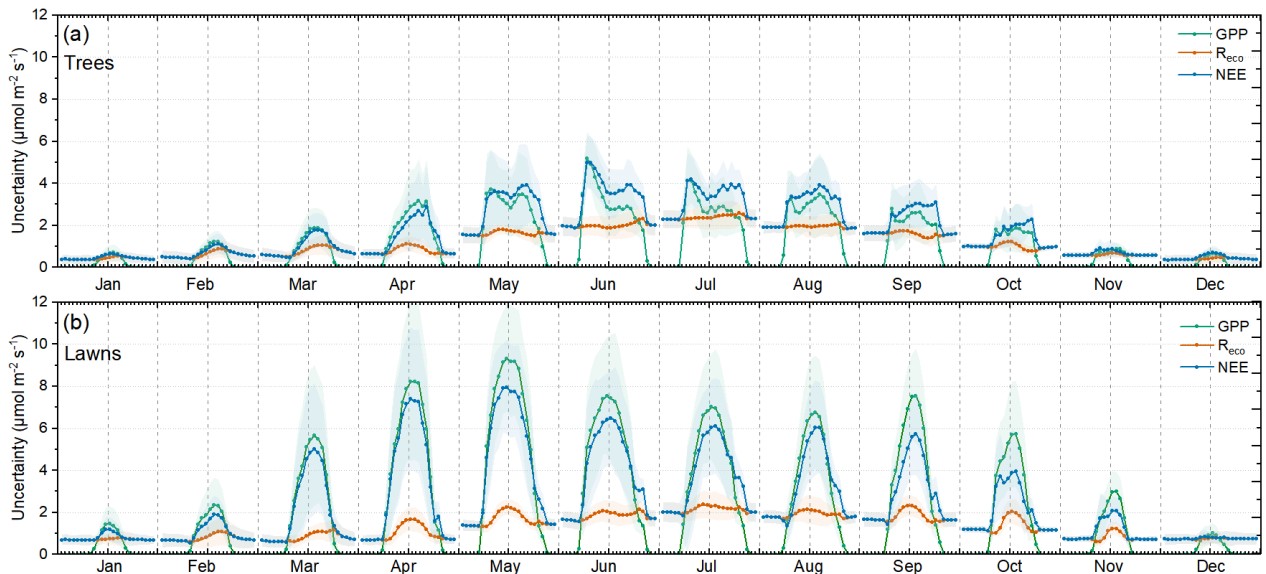

**Figure 7. Monthly diurnal hourly averages (dots) and standard deviations (shading) of the estimated GPP, $R_{eco}$ and NEE uncertainties, for (a) park trees and (b) lawns. The monthly averages consider the years 2022 and 2023 together.**

### 3.5. Share of biogenic fluxes in the city

The comparison between the monthly biogenic fluxes, as well as their uncertainties, to the total anthropogenic emissions is presented in Table 3. It is evident that during wintertime the biogenic fluxes and their uncertainties were insignificant compared to the anthropogenic emissions. However, during late spring, the anthropogenic emissions abruptly decreased due to the absence of building heating and at the same time the biogenic components got more active. Therefore, GPP and $R_{eco}$ are the competing processes whose magnitudes reached 30 to 36 % of the net monthly $CO_2$ balance when considered

individually (Table 3). The contribution of NEE peaked during May, when both trees and lawns were in their optimal season, reaching -10.8 % of the monthly net budget. Even though the contribution was rather small on a monthly scale due to the trade-off between GPP and $R_{eco}$, its estimated uncertainty was similar to the one of GPP (Table 3). It appears that the uncertainties associated with NEE can be an important part of the urban $CO_2$ cycle even in a city centre during summer and early autumn periods, their magnitudes found in this study to range between 6.1−10.2 % of the net monthly balance. It is

important to note that the simple method of estimating the uncertainties of the biogenic flux components from the model spread may underestimate the real uncertainties as the models partially rely on the same inputs and on similar assumptions. This is shown, for example, by the fact that the model spread does not always encompass the observed values of $R_{eco}$ at the lawn locations.

**Table 3. Monthly average fluxes ($\mu$mol $CO_2$ m$^{-2}$ s$^{-1}$) of the city central 2 km x 2 km domain (Fig. 1), considering the total anthropogenic emissions (building heating, industry, vehicle traffic, human respiration) and the biogenic $CO_2$ flux components (GPP, $R_{eco}$, NEE) and uncertainties (unc., Section 3.3). The percentages (%) to the total monthly $CO_2$ balances are given in parentheses. The data are monthly averages of the years 2022 and 2023.**

|     | Total anthr. | GPP | | GPP unc. | | $R_{eco}$ | | $R_{eco}$ unc. | | NEE | | NEE unc. | |
| --- | --- | --- | --- | --- | --- | --- | --- | --- | --- | --- | --- | --- | --- |
| Jan | 29.1 | 0.1 | (0.4) | 0.1 | (0.3) | 0.6 | (1.9) | 0.2 | (0.6) | 0.4 | (1.5) | 0.2 | (0.6) |
| Feb | 27.0 | 0.3 | (1.1) | 0.2 | (0.6) | 0.6 | (2.2) | 0.2 | (0.6) | 0.3 | (1.1) | 0.2 | (0.8) |
| Mar | 21.5 | 0.9 | (4.0) | 0.5 | (2.2) | 0.9 | (4.0) | 0.3 | (1.2) | 0.0 | (0.0) | 0.5 | (2.2) |
| Apr | 18.4 | 1.8 | (9.9) | 0.8 | (4.4) | 1.2 | (6.9) | 0.3 | (1.6) | -0.5 | (-3.1) | 0.8 | (4.2) |
| May | 9.3 | 2.9 | (34.2) | 0.9 | (10.2) | 2.0 | (23.3) | 0.4 | (4.5) | -0.9 | (-10.8) | 0.8 | (9.5) |
| Jun | 8.7 | 2.9 | (36.0) | 0.8 | (9.4) | 2.4 | (29.9) | 0.3 | (4.3) | -0.5 | (-6.1) | 0.8 | (10.2) |
| Jul | 8.7 | 2.5 | (28.6) | 0.7 | (8.2) | 2.6 | (30.1) | 0.3 | (3.7) | 0.1 | (1.6) | 0.8 | (9.2) |
| Aug | 8.7 | 1.9 | (20.7) | 0.6 | (7.0) | 2.5 | (27.1) | 0.4 | (4.1) | 0.6 | (6.4) | 0.7 | (7.8) |
| Sep | 9.6 | 1.9 | (19.6) | 0.5 | (5.3) | 2.1 | (21.6) | 0.4 | (4.5) | 0.2 | (2.1) | 0.6 | (6.1) |
| Oct | 10.2 | 1.2 | (11.2) | 0.4 | (3.9) | 1.6 | (15.5) | 0.1 | (1.3) | 0.5 | (4.3) | 0.4 | (3.9) |
| Nov | 23.0 | 0.5 | (2.2) | 0.2 | (1.0) | 1.1 | (4.5) | 0.2 | (0.8) | 0.5 | (2.3) | 0.3 | (1.2) |
| Dec | 29.0 | 0.1 | (0.4) | 0.1 | (0.3) | 0.6 | (2.1) | 0.2 | (0.6) | 0.5 | (1.7) | 0.2 | (0.6) |

## 4. Discussion

### 4.1 Advantages and disadvantages of different types of biogenic $CO_2$ exchange models in urban contexts

Even though the models tested were forced with the same data, it is challenging to establish a clear hierarchy of model performance using the available measurements, as the performance of individual models seems to differ with different drivers or proxies for $CO_2$ exchange (Fig. 6). All tested models agreed on the timing of the spring recovery of trees in terms of GPP in 2022. However, at the beginning of the season 2023, SUEWS, VPRM, and diFUME were again consistent among each other but their GPP values started to grow earlier than the observed recovery in sap flow and LAI (Fig. 3a,b, B1a). JSBACH simulated an even earlier onset. On the other hand, the seasonal dynamics of JSBACH was found to be accurate in a recent study in hemiboreal tree-covered urban areas in Helsinki (Thölix et al., 2024), suggesting that the timing of spring recovery dynamics in JSBACH should be examined with longer and wider datasets at different ecosystem types. The overall evaluation of modelled tree GPP based on the sap flow dynamics indicates that all tested models provide reasonably accurate GPP estimates. The $R_{eco}$ for tree-covered areas by JSBACH, SUEWS and VPRM were closer to each other than that of diFUME (Fig. 3c,d). Unfortunately, especially the autotrophic respiration of the aboveground tree parts is difficult to measure and therefore, the analysis is unable to reveal the best model regarding tree $R_{eco}$.

For lawns, JSBACH successfully estimated soil moisture and outperformed other models in predicting observed $R_{eco}$, which on the other hand, exhibited much more variability than the simulations (Fig. 4c). Even though JSBACH simulated slightly higher diurnal variation in $R_{eco}$, the dynamics of the simple VPRM and the process-based ecosystem model JSBACH are comparable (Fig. 4c,d). Compared to VPRM and JSBACH, diFUME and SUEWS simulated higher and lower $R_{eco}$ respectively, and consistently lower GPP. Despite the dry periods in the summer months, VPRM and JSBACH also predicted similar annual cycles and day-to-day variability of GPP but unfortunately, direct measurements of lawn GPP were not available in this study. Nonetheless, GPP estimates by JSBACH for lawn has been tested and shown to be sufficient in hemiboreal zone using repeated measurements for photosynthesis (Trémeau et al., 2024). VPRM was evaluated in an urban forest and grassland area using field-based reference data by Winbourne et al. (2022), who found that $R_{eco}$ was substantially overestimated during winter months but the GPP was approximated with better accuracy. This is consistent with the present study, which shows higher $R_{eco}$ values from VPRM compared to the other models in winter.

Overall, the four models are different in terms of sophistication and structure, as well as in their using different data streams. Therefore, different advantages and disadvantages can be expected, depending on the application and context. All the models require temperature and radiation inputs. However, VPRM requires the least number of data streams among the models (Table 2). diFUME and VPRM use satellite data to track vegetation phenology (Table A1), making them more suitable for monitoring purposes, especially in urban areas where the vegetation type heterogeneity is so pronounced that it is very difficult to model accurately. As demonstrated also by the current study, these satellite-based models are able to detect

changes in leaf area due to phenological shifts or stressful events, such as drought, providing valuable insights into ecosystem responses to environmental change. However, these models are not able to predict future carbon cycling under different climate scenarios or planning strategies, as they rely heavily on observations.

Despite the different levels of sophistication of the four models on simulating GPP, all consider some sort of hyperbolic function to model the response of gross photosynthesis to light and a bell-shaped dependence on air temperature (Table A1). On the other hand, the responses of photosynthesis to vapour pressure deficit and soil water availability are accounted very differently in each model. VPRM uses a very simplistic approximation of drought effects on GPP based on LSWI, diFUME and SUEWS use empirical functions, while JSBACH has the most sophisticated approach to modelling drought effects (Section 2.3, Table A1). The generally good agreement between the models in simulated tree GPP seasonally and diurnally indicates that the simple process approximations were sufficient, possibly due to the lack of intense drought effects on tree GPP in this study based on the sap flow observations (Fig. 3a,b). On the other hand, drought effects on lawn GPP were captured by all models during August 2022, with VPRM showing the most intense drought-induced GPP reductions which were repeated in summer 2023 (Fig. 4a,b). These findings indicate that VPRM is very sensitive to LSWI and EVI indices (Fig. B1b,c), which can drive GPP to excessively low values during dry periods, in contrast to the process-based JSBACH and the empirical functions of diFUME and SUEWS. Unfortunately, we do not have any independent measurements of lawn GPP in this study to evaluate which type of model is closest to the truth.

Even though there are some similarities between the models in the representation of the GPP process, the description of $R_{eco}$ significantly differs in terms of approximations and sophistication. JSBACH is the only model tested that includes carbon pools, which are essential for studying long-term temporal dynamics. This feature allows JSBACH to model the behaviour of soil carbon pools and their changes over extended periods. For instance, if high decomposition rates persist, the decreasing soil carbon pool also decreases heterotrophic emissions, whereas other models tested do not have such feedbacks and use only empirical environmental response functions. diFUME uses a more detailed approach compared to SUEWS and VPRM, separating above and below ground respiration and using $T_{air}$, $T_{soil}$, SWC and LAI as proxies, whereas SUEWS and VPRM use an exponential and a linear response function on $T_{air}$ respectively (Table A1). Despite the differences in the representation of $R_{eco}$, the seasonal and diurnal variabilities were not very different between the four models (Fig. 3c,d, 4c,d) and the differences detected were not clearly related to the level of sophistication but rather to the choice in parametrisation. For example, diFUME tree $R_{eco}$ was higher than the other models because the parameterisation was kept the same for lawn and tree sites. JSBACH showed the best performance in predicting lawn $R_{eco}$ (Fig. 6d) but on the other hand, ecosystem process models such as JSBACH require very detailed input information (e.g. on soil carbon stocks) and are based on full-cycle assumptions that are difficult to meet in highly managed and disturbed urban ecosystems where carbon pools are constantly altered by human interventions (Golubiewski, 2006). Furthermore, process-based models such as JSBACH require a large number of input parameters compared to simple light-use-efficiency models such as VPRM which has, in some cases, been found to outperform process-based models in explaining $CO_2$ variability (Gourdji et al., 2022).

It is important to highlight the limitations of the current study in terms of evaluating the models in the urban context. This study focuses on four urban parks where the effects of buildings and paved surfaces on energy and water exchange are not as pronounced as in urban canyons. Urban canyon vegetation and street trees in particular grow in a very different environment compared to urban parks (Dahlhausen et al., 2018; Nielsen et al., 2007) and their ecophysiological responses and dynamics can be potentially different from park trees. In this study, we apply SUEWS and diFUME models, which are specifically designed for urban applications and include aspects of urban morphology in the simulations of biogenic $CO_2$ fluxes. Specifically, SUEWS stands out in assessing the urban energy balance and its effect on local air temperatures. This model is particularly advantageous in urban contexts where a detailed understanding of the energy exchanges is crucial for microclimate. A distinct feature of diFUME is that it has a 3D radiation module that takes into account building shading and diffuse radiation extinction within urban canyons. As the role of buildings, runoff and local temperature are not as evident in the park applications of the current study, it is possible that the additional benefits of these models are not highlighted in the model intercomparison. In order to achieve a more thorough evaluation of the models and their applicability in the urban context, future research is needed focusing on different urban environments, including vegetated street canyons.

### 4.2 In-situ observations and challenges

Despite the technical, logistical and methodological challenges in measuring sap flow and estimating tree transpiration in urban areas, such observations can be very valuable for monitoring tree physiological dynamics especially in areas of extreme heterogeneity where other techniques, such as eddy covariance, cannot be applied (Ewers and Oren, 2000; Peters et al., 2010). The sap flux densities measured in this study were very consistent with the seasonal and diurnal variability and magnitude for similar species, presented in other studies in suburban and urban areas (Bush et al., 2008; Peters et al., 2010; Rahman et al., 2017). *Tilia* sp. and *Platanus* sp. trees measured in this study, being semi-ring porous species (Schoch et al., 2004), present relatively constant maximum sap flux densities throughout the growing period due to the increased stomatal regulation (Bush et al., 2008; Peters et al., 2010). This suggests limited stomatal opening during high VPD and incoming radiation conditions, which also limits photosynthetic rates. Similar to the results of the present study, measured sap flux densities in other studies responded only marginally to decreases in topsoil water content during summer (Peters et al., 2010; Rahman et al., 2017), indicating that the water availability in deeper soil layers is more important for maintaining the required hydraulic conductance of mature trees during summer. In the present study, we used a simple linear regression approach to derive reference daily tree GPP values from sap flow observations rather than a more sophisticated approach (e.g. Schäfer et al., 2003) to avoid involving a lot of modelling in our reference dataset. The disadvantage of the used approach is that it cannot predict the absolute magnitude of the GPP and only provides information about the daily variability.

Flux chamber $R_{soil}$ and lawn $R_{eco}$ observations serve as a direct reference for model evaluation, but they can greatly vary across different urban locations and there are technical and logistical limitations in their application across urban spaces. For example, this study focused on large green areas in Zurich to obtain representative measurements of $CO_2$ flux behaviour of

the park lawns under the city's management practices. These lawns are very busy public areas, therefore long-term installations (e.g. automated chambers) are not possible and the observations are limited to specific dates and times of field campaigns. The wealth of information that can be derived from continuous long-term automated chambers (e.g. Hill et al., 2021) cannot be compared to the sporadic field measurements. For example, the sudden increase of $R_{eco}$ in the sunny lawn of Hardaupark in April 2023 would probably be easier to confirm and explain if we had continuous observations. Moreover, clear chamber measurements over park lawns were not implemented during this study but would be considered very helpful for estimating lawn NEE and partitioning to GPP and $R_{eco}$. This information during the field campaigns would be valuable for the complete evaluation of the model performance over lawns. Such measurements are currently implemented in the ICOS-Cities project pilot case study sites of Paris and Munich complementing the observations presented in this study.

The observations of $R_{soil}$ and lawn $R_{eco}$ presented in this study are comparable with what has been reported in the literature concerning urban park lawns. Even though it is well established that the main drivers of $R_{soil}$ and $R_{eco}$ is $T_{soil}$ and water content (e.g. Raich and Schlesinger, 1992), the response functions and magnitudes of respiration in urban parks can be significantly different from native grasslands (Kaye et al., 2005). There are several studies that have measured soil organic carbon (SOC) pools across urban parks and natural ecosystems (e.g. Golubiewski, 2006; Kaye et al., 2005; Pouyat et al., 2006; Weissert et al., 2016). SOC pools vary across urban parks and are an important determinant for $R_{soil}$ as well as plant productivity (Golubiewski, 2006; Kaye et al., 2005). In general, studies suggest that sequestration of soil C in parks dominated with grass increases linearly with park age (Lal and Augustin, 2012), reaching even higher biomass productivity and larger SOC pools than native grasslands (Golubiewski, 2006; Kaye et al., 2005; Pouyat et al., 2006; Weissert et al., 2016). However, this process can be severely affected by management practices, such as mowing, fertilisation and irrigation, which can also affect the balance between respiration and photosynthetic rates.

Allaire et al. (2008) compared respiration rates of urban turfgrass, measured with chambers under different fertilisation and mowing treatments, and found that mowing frequency had higher impact on respiration than fertilisation and soil characteristics. Similar to our observations, the lawn $R_{eco}$ observed by Allaire et al. (2008) peaked suddenly in spring in the treatment with frequent mowing, reaching maximum at around 14 µmol m$^{-2}$ s$^{-1}$. Kaye et al. (2005) found consistently high $R_{soil}$ and belowground C allocation in intensively mowed and irrigated urban lawns, 2.5 times higher than in adjacent native and agricultural ecosystems. It appears that mowing with return of clippings on the ground increases rapidly the $R_{eco}$ under wet and warm conditions due to accelerated microbial activity. Furthermore, mowing alters the microenvironment of the lawn surface, leading to increased radiation exposure and temperatures of grass leaves, which can significantly alter leaf respiration rates (Shen et al., 2013). The $R_{eco}$ observations in this study during the spring and summer of 2023 were probably affected by mowing, which increased $R_{eco}$ to very high values in sunny areas, even higher than 20 µmol m$^{-2}$ s$^{-1}$ in some cases. Detailed information on the management practices and their timing would be very valuable for interpreting the observations but unfortunately it was not available. High values of maximum grass $R_{eco}$, similar to the present study and Allaire et al. (2008), were also reported by Liss et al. (2009) and Weissert et al. (2016) over park, residential and sportsfield lawns under warm and moist conditions. However, the mowing effects were not clear in these studies.

**4.3 Possibilities to mitigate climate change via urban C sequestration**

Vegetation is able to sequester part of the anthropogenic emissions originating from fossil fuel burning. In this study, the vegetation uptake in central Zurich covered nearly 11 % of the total carbon budget in the optimal month of May. The summertime offset in this study is smaller than observed in Helsinki (42 %, Havu et al., 2024), New York (20-40 %, Wei et al., 2022) or Florence (26 %; Vaccari et al., 2013), which is likely due to the fact that the studied area is highly built with only 30% of vegetation. The forest-covered hills surrounding the city, for example, are not included in this area. As photosynthesis is minimal during the winter months, this results in a small offset of anthropogenic emissions on an annual scale. In central Helsinki, vegetation was found to uptake 3% of the anthropogenic emissions on an annual scale whereas at the scale of the whole city the offset was 7% (Havu et al., 2024; Järvi et al., 2019). Prior studies have reported annual offsets of 2.1% in Boston (Hardiman et al., 2017) and 6.2% in Florence (Vaccari et al., 2013) but naturally the offset is strongly dependent on the strength of anthropogenic emissions.

These numbers demonstrate the potential of urban vegetation to mitigate climate change. With the anthropogenic emissions potentially decreasing in the future, the contribution of biogenic fluxes will increase. It is also important to remember that the C sequestration of vegetation is highly dependent on its well-being and thus it is critical to find natural solutions that are resilient to heat, drought and pests. Species diversity plays a crucial role here.

**5. Conclusions**

This study compared for the first time four different types of biosphere models (diFUME, JSBACH, SUEWS, VPRM) in terms of their accuracy in simulating $CO_2$ exchange (GPP, $R_{eco}$, NEE) in urban parks. Model simulations were evaluated against in-situ observations, in order to identify advantages and disadvantages of the different model types. The results showed a good agreement between all models in terms of magnitude and seasonality of park tree GPP, but there is potentially a challenge for process-based models (JSBACH, SUEWS) to accurately simulate the onset of tree greening in spring compared to remote sensing based models (diFUME, VPRM). On the other hand, there were larger differences between the $R_{eco}$ simulations over both trees and lawns, as well as lawn GPP. However, in-situ observations showed that all models underestimated the high lawn $R_{eco}$ values in spring and summer, with the effects of grass mowing most likely being a factor that could not be captured by any model. The annual NEE estimates of the models agreed that on average park lawns acted as $CO_2$ sources and tree-covered park areas as $CO_2$ sinks during the simulation period with the exception of the diFUME model which simulated both trees and lawns as $CO_2$ sources. When compared to the anthropogenic $CO_2$ emissions, the simulated monthly NEE and its uncertainties were a considerable fraction of the net urban emissions during the vegetation growing period, reaching its maximum impact in late spring with NEE being 10.8 % and its uncertainty 10.2 % of the net $CO_2$ balance.

Overall, the results of this study do not provide a clear indication of the superiority of sophisticated process-based models, such as JSBACH, over simple light use efficiency models, such as VPRM, in simulating the dynamics of $CO_2$ fluxes in

urban parks. However, this study is restricted to urban park vegetation and the presented results cannot be easily generalised to other urban environments, such as vegetated urban canyons. Further research is needed to get a full understanding of the model applicability and efficiency across the wide heterogeneity of urban areas. The main limitation towards a more thorough model evaluation is the lack of urban in-situ observations, mainly due to the technical, methodological and logistical challenges of deriving representative observations across urban vegetation. More detailed, frequent and spatially extended in-situ observations would be necessary for a full assessment of model applicability. Specifically, grass chamber observations with clear and opaque chambers and even long-term continuous installations would be ideal to evaluate the model performance to simulate park lawn GPP, $R_{eco}$ and NEE. However, precautions must be taken in the design and implementations of such observations to avoid changing the microenvironment of the observation site and to allow the normal management procedures to be applied (e.g. mowing), so that the observation sites would still be representative of the park conditions. Regarding $CO_2$ exchange at tree scale, sap flow observations are a promising method to monitor tree physiological behaviour despite the application challenges. The extreme heterogeneity of tree species, age and environmental conditions of the urban areas remains a huge challenge for the urban model evaluation studies.

## Appendix A. Model architecture and process representation

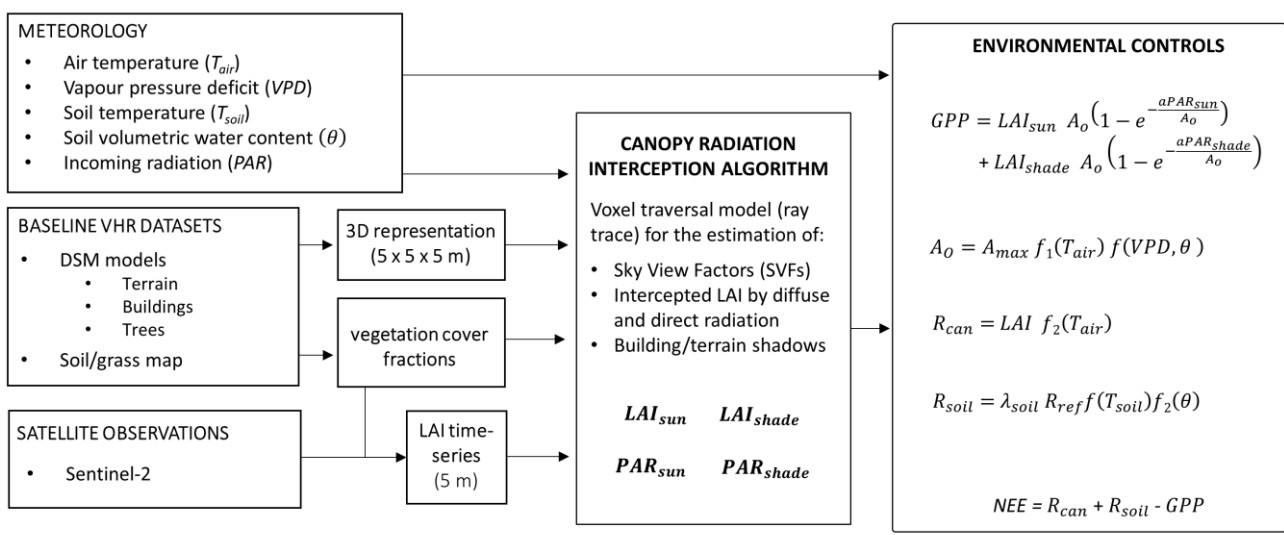

**Figure A1. Process flow diagram of the diFUME model. The detailed description of the model can be found in Stagakis et al. (2023a).**

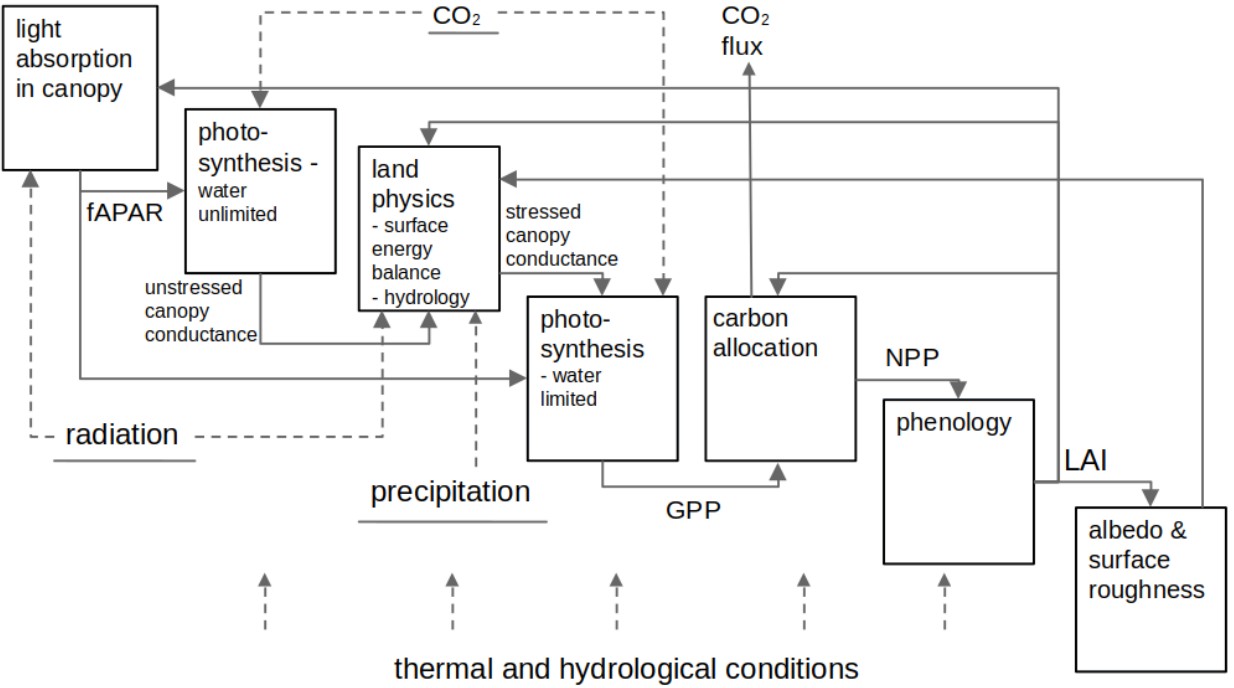

**Figure A2. Process flow diagram of the JSBACH model. The detailed description of the model can be found in Reick et al. (2013).**

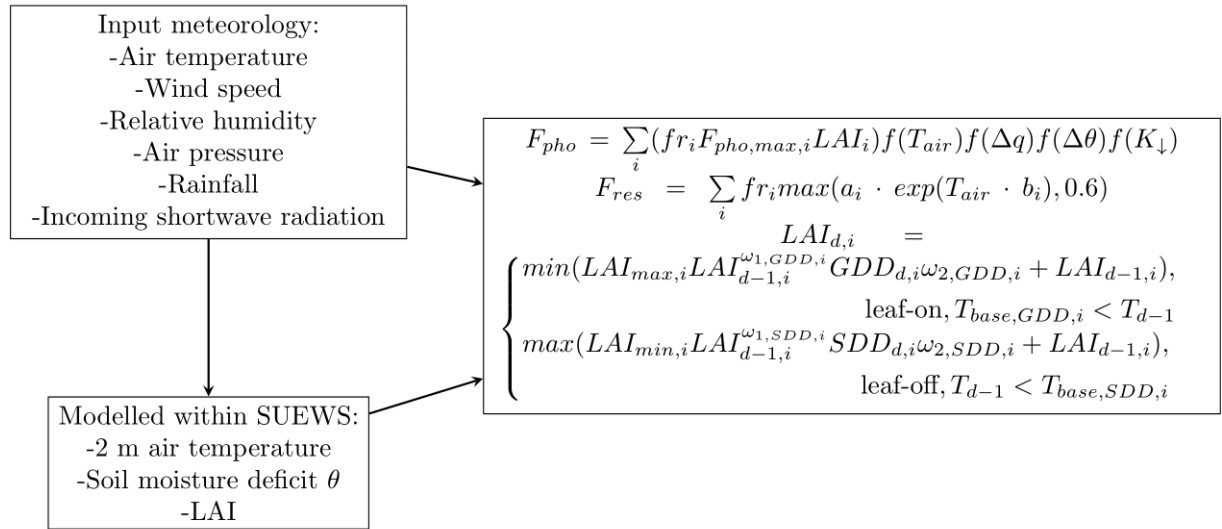

Figure A3. Process flow diagram of the SUEWS model. The detailed description of the model can be found in Järvi et al. (2011, 2019).

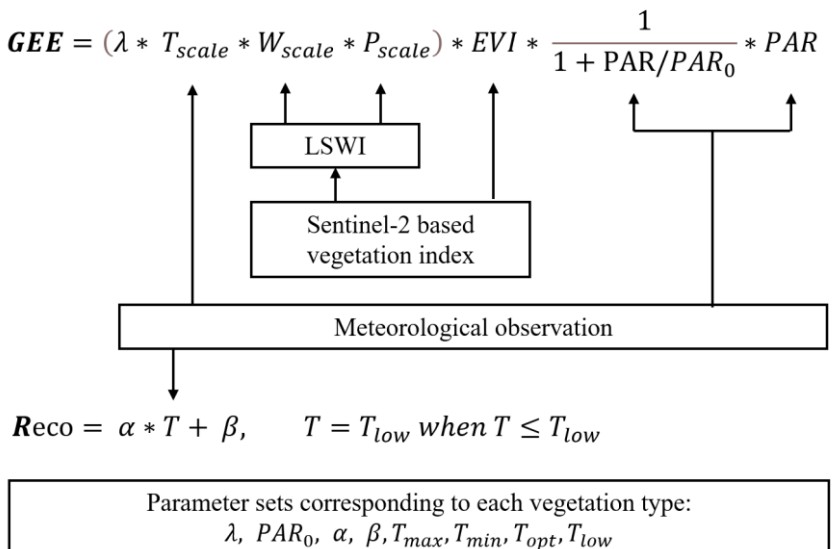

Figure A4. Process flow diagram of the VPRM model. The detailed description of the model can be found in Mahadevan et al. (2008).

Table A1. Overview of the main equations used by each model to estimate gross primary production (GPP), ecosystem respiration (R$_{eco}$) and phenology. The original notation of each model was preserved to facilitate the comparison with the relevant publications and the flow diagrams. The parameters of the equations are explained in the Supplement. The detailed descriptions of each model can be found in the respective references provided in Section 2.3.

| | GPP | R$_{eco}$ | Phenology |
|---|---|---|---|
| diFUME | $GPP = \sum_{l=1}^{n} \left[ \begin{array}{l} LAI_{sun,i} A_o (1 - exp(-aPAR_{sun,i}/A_o)) + \\ LAI_{shade,i} A_o (1 - exp(-aPAR_{shade,i}/A_o)) \end{array} \right]$ <br><br> $A_o = A_{max} \cdot f(T_{air}) f(VPD, \theta)$ <br><br> $f(T_{air}) = exp\left[ -(T_{air} - T_{opt})^2 / 2W^2 \right]$ <br><br> $f(VPD, \theta)$ based on: $g_s = \left( g_o + a_1 \frac{A_{net} \, f(T_{air})}{\left(1 + \frac{VPD}{D_o}\right)(c_s - \Gamma)} \right) \left( 1 - \frac{(\theta_{ref} - \theta)^{b_1}}{(\theta_{ref} - \theta_g)^{b_1}} \right)$ | $R_{eco} = LAI \, D_{sc} R_l +$ <br><br> $\lambda_{soil} R_{S,ref} \, exp\left[ E_0 \left( \frac{1}{T_{ref,S} - T_0} - \frac{1}{T_{soil} - T_0} \right) \right] \left[ \frac{(\theta - \theta_0)^b}{(\theta_{ref} - \theta_0)^b} \right]$ <br><br> $R_l = R_{l,ref} Q_{10}^{\frac{T_{air} - T_{ref,l}}{10}}$ | *LAI derived by Copernicus High Resolution Vegetation Phenology and Productivity (HR-VPP) product* |
| JSBACH | *Due to the complexity of the full Farquhar et al. (1980) model implementation in JSBACH, only an outline is provided here.* <br><br> $GPP = LAI \, A_{stress}$ <br><br> $A_{stress} = min(J_{C,stress}, J_{E,stress}) - r_d$ <br><br> $g_{L,stress}^{H_2O} = f_{ws} \, g_L^{H_2O}$ | $R_{eco} = R_m + R_g + R_h$ <br><br> $R_m = LAI r_d / f_{leaf}$ <br><br> $R_g = (CC - 1)NPP, \text{ when } NPP>0$ <br><br> $CC = (NPP + R_g)/NPP$ <br><br> $R_h = \left(1 - f_{faeces}\right) F_{grazing} + k_h C_h + F_{soil}$ | $\frac{dLAI}{dt} = kLAI\left(\frac{LAI}{LAI_{max}}\right) - pLAI$ |
| SUEWS | $F_{pho} = \sum_i (fr_i F_{pho,max,i} LAI_i) g(T_{air}) g(\Delta q) g(\Delta \theta) g(K_\downarrow)$ <br><br> $g(T_{air}) = [(T_{air} - T_L)(T_H - T_{air})^{T_C}]/[(G_5 - T_L)(T_H - G_5)^{T_C}]$ <br><br> $T_C = (T_H - G_5)/(G_5 - T_L)$ <br><br> $g(\Delta q) = G_3 + (1 - G_3)G_4^{\Delta q}$ <br><br> $g(\Delta \theta) = [1 - exp(G_6(\Delta \theta - \Delta \theta_{WP}))]/[1 - exp(-G_6 \Delta \theta_{WP})]$ <br><br> $g(K_\downarrow) = [K_\downarrow/(G_2 + K_\downarrow)]/[K_{\downarrow max}/(G_2 + K_{\downarrow max})]$ | $F_{res} = \sum_i fr_i \, max(a_i exp(T_{air} b_i), 0.6)$ | $LAI_{d,i}$ <br> $= \begin{cases} min(LAI_{max,i}, LAI_{d-1,i}^{\omega_{1,GDD,i}} GDD_{d,i} \omega_{2,GDD,i} + LAI_{d-1,i}), \\ \quad leaf-on, T_{base,GDD,i} < T_{d-1} \\ max(LAI_{min,i}, LAI_{d-1,i}^{\omega_{1,SDD,i}} SDD_{d,i} \omega_{2,SDD,i} + LAI_{d-1,i}), \\ \quad leaf-off, T_{d-1} < T_{base,SDD,i} \end{cases}$ |
| VPRM | $GEE = (\lambda T_{scale} W_{scale} P_{scale}) EVI [1/(1 + PAR/PAR_0)] PAR$ <br><br> $T_{scale} = (T - T_{min})(T - T_{max})/\left[(T - T_{min})(T - T_{max}) - (T - T_{opt})^2\right]$ <br><br> $W_{scale} = 1 + LSWI/(1 + LSWI_{max})$ <br><br> $P_{scale} = (1 + LSWI)/2$ | $Reco = \alpha T + \beta$ <br><br> $T = T_{low}, if \, T \le T_{low}$ | *Sentinel-2 vegetation indices* |

**Appendix B. Comparison between in-situ observations and modelled parameters**

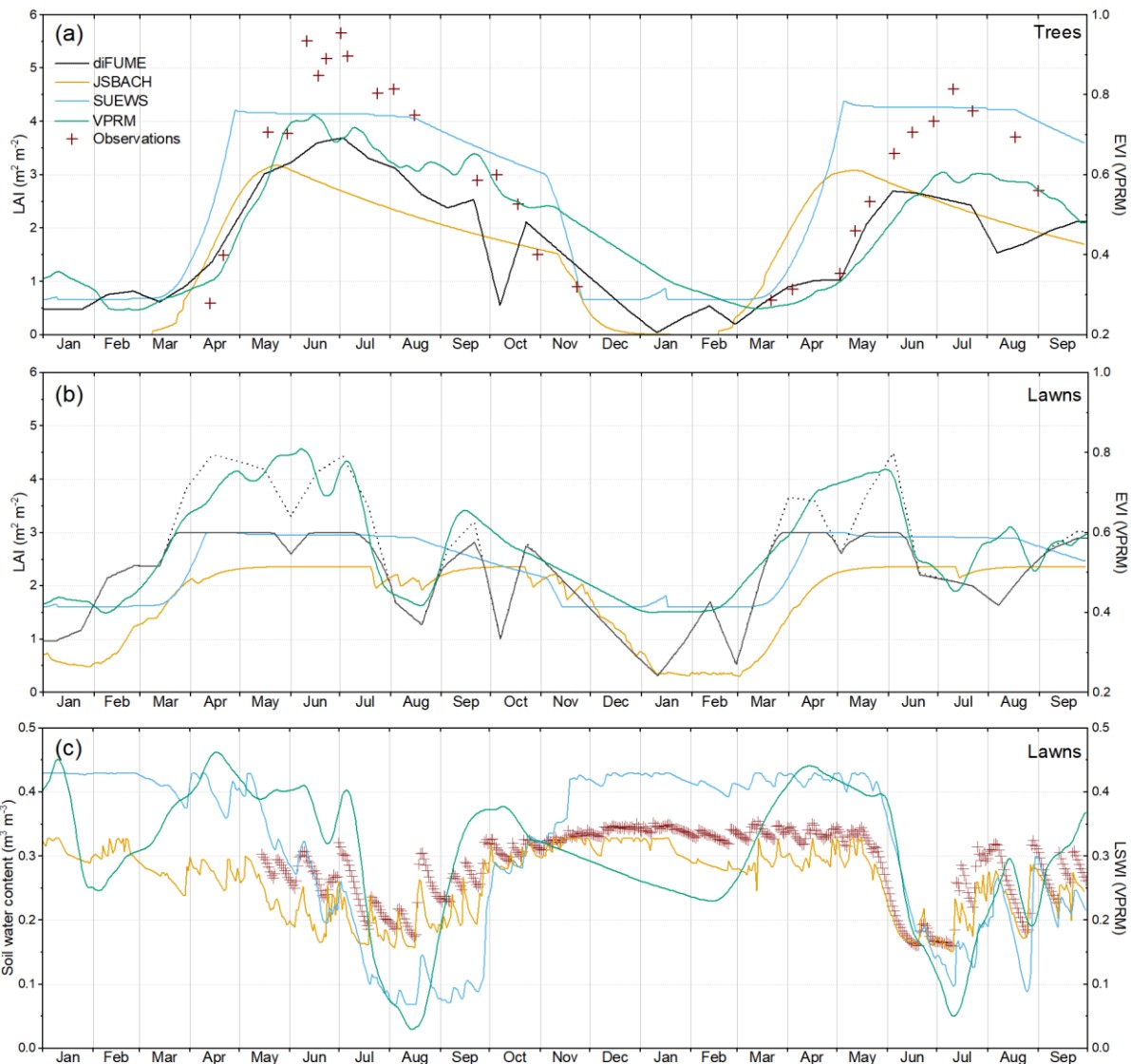

**Figure B1.** Time-series of daily (a) modelled and observed tree leaf area index (LAI) by diFUME, JSBACH and SUEWS together with EVI estimates by VPRM, (b) lawn LAI by diFUME, JSBACH and SUEWS and VPRM lawn EVI, (c) observed and modelled soil water content in the lawns by diFUME, JSBACH and SUEWS together with Land Surface Water Index (LSWI) by VPRM. In panel b, dotted line shows the original diFUME LAI before the upper threshold of 3 m$^2$ m$^{-2}$. All datasets are averaged between the four parks, except tree LAI where only Bulingerfof and Hardaupark are used.

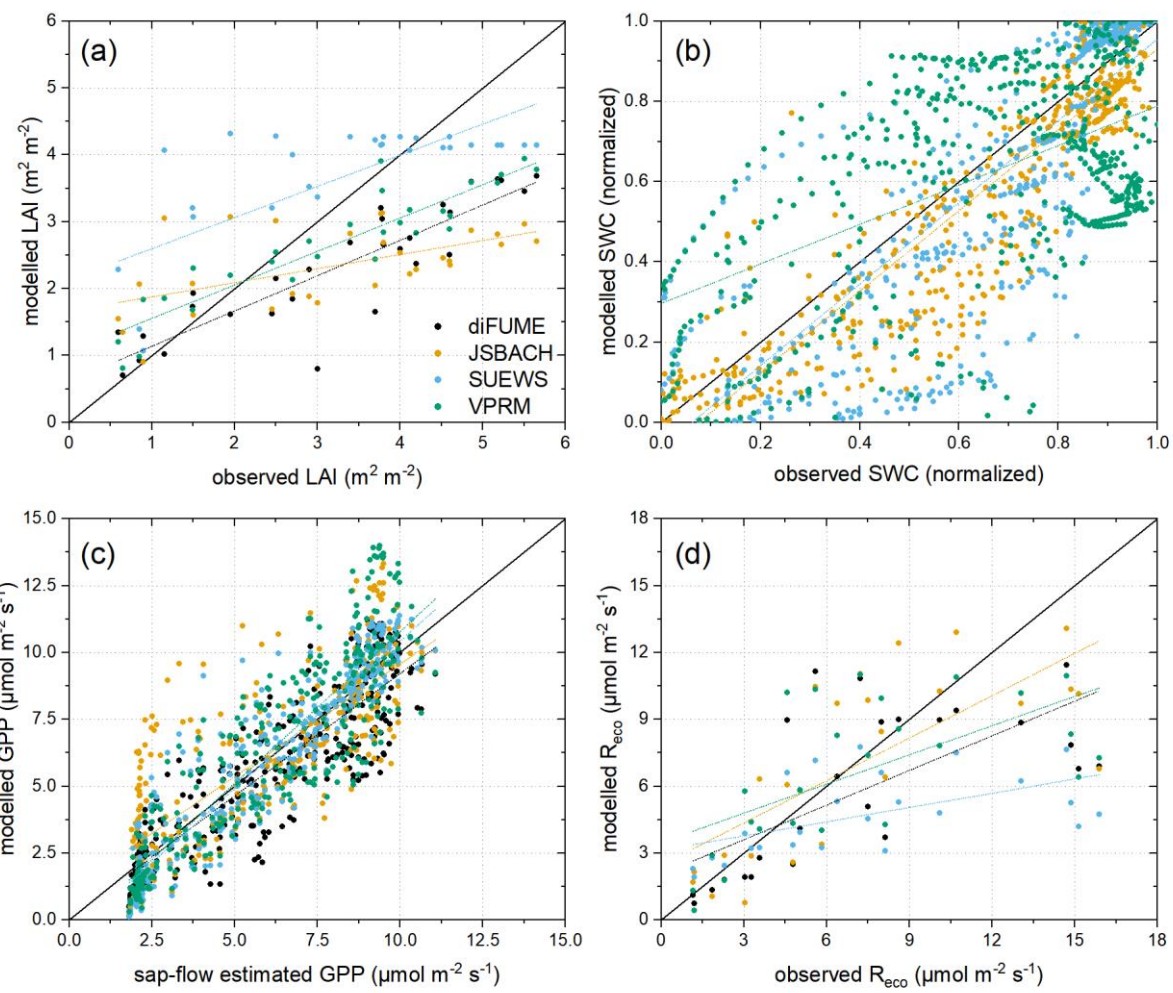

**Figure B2. Scatterplots between (a) observed and modelled tree leaf area index (LAI, b) normalised observed and modelled soil water content (SWC) of lawns, (c) observed and modelled tree gross primary production (GPP), and (d) observed and modelled lawn ecosystem respiration ($R_{eco}$). The 1:1 lines are thick black lines and the linear fits for each model are presented with the respective colour. The evaluation metrics of the comparisons are presented in the Taylor diagram (Fig. 6). For the conversion of**
835 **VPRM EVI to LAI, VPRM LSWI to SWC and tree sap flow observations to tree GPP, see Section 2.5.**

**Data availability**

All datasets produced in the present study:  i. in-situ observations, ii. land cover map, iii. meteorology, iv. modelled $CO_2$ fluxes and model parameter sets, are available in Zenodo repository under Creative Commons Licence CC-BY-4.0 (https://doi.org/10.5281/zenodo.13222637).

**Author contribution**

SS: Conceptualization; Data curation; Formal analysis; Investigation; Methodology; Validation; Visualization; Writing - original draft. DB: Methodology; Conceptualization; Visualization; Writing – review & editing. JL: Formal analysis; Investigation; Writing - original draft. LB: Conceptualization; Investigation; Validation; Visualization; Writing – review & editing. AK: Data curation; Investigation; Validation; Visualization; Writing – review & editing. LC: Formal analysis; Investigation; Writing - original draft. LJ: Conceptualization; Funding acquisition; Project administration; Conceptualization; Writing - original draft preparation. MH: Conceptualization; Investigation; Writing – review & editing, JC: Conceptualization; Funding acquisition; Project administration, Resources, Writing – review & editing. SE: Investigation; Writing – review & editing. LK: Conceptualization; Funding acquisition; Project administration; Writing – original draft preparation; Writing – review & editing.

**Competing interests**

The authors declare that they have no conflict of interest.

**Acknowledgments**

This study has received funding from the European Union's Horizon 2020 research and innovation programme under grant agreement No 101037319 in the framework of ICOS Cities project (Pilot Applications in Urban Landscapes - Towards integrated city observatories for greenhouse gases - PAUL). We acknowledge also the funding by the Research Council of Finland (grants #321527, 325549), the Strategic Research Council working under the Research Council of Finland (grants #335201, 335204), and ACCC Flagship program (grants #337552, 337549). Financial support from ICOS Switzerland (ICOS-CH) Phase 3 (Swiss National Science Foundation, grant 20F120_198227) is also acknowledged. ICOS Ecosystem Thematic Centre and University of Antwerp are acknowledged for providing the SunScan instrument used for the LAI field observations. Grün Stadt Zürich is acknowledged for giving permission and assisting with the field measurements. Two anonymous Reviewers are acknowledged for their insightful comments and suggestions that helped the improvement of the manuscript.

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
