# Peer review of "Intercomparison of biogenic CO2 flux models in four urban parks in the city of Zurich"

_EGUsphere, 2024_

## Author Response (AR2)

**Response to Anonymous Referee #1**

We would like to sincerely thank Referee #1 for their time in reviewing this manuscript and for their positive feedback. We found the provided comments and suggestions very constructive for improving this paper. Our responses to each comment are provided below in black fonts and the Referee's comments are indicated in blue fonts. All the mentioned line and section numbers refer to the originally submitted manuscript unless stated otherwise.

Major Comments:

Model Architecture and Process Representation

The manuscript would benefit from:
- A more comprehensive comparison of core model mechanisms
- Clear explanation of how each model transforms inputs into biogenic flux outputs
- Detailed coverage of key process differences, particularly:
  - GPP calculations from radiation and vegetation parameters
  - Soil moisture influences on photosynthesis and respiration
  - Temperature effects on respiration
  - Phenology implementation approaches

We agree that a more detailed and comprehensive description of the models would help the readers understand better their differences and would promote a better interpretation and discussion of the results. We have followed the Referee's suggestions and we have included process flow diagrams for each model and a Table that summarizes the key equations used by each model for the calculation of GPP, $R_{eco}$ and phenology. We have added this new information in a new Appendix and we have updated Section 2.3 (Model description) with more details, explaining better the different methodological approaches followed by each model, referring to the new information supplied in the Appendix. Additionally, we added four Tables in a Supplement document, which provide the description and the values of the parameters used by each model in this paper. These Tables complement the information provided in the new Appendix, facilitating the understanding of the different model approaches and their parametrisation in this paper.

Below you can find the new text added in Section 2.3 (model description):

[revised manuscript text omitted]

To facilitate this comparison, please include:
- Process flow diagrams for each model
- Comparative table of key equations and methodological approaches

The process flow diagrams of each model and the table of key equations have been added in a new Appendix. The new information is presented below. The Tables with the model parameters are provided in the Supplement file (attached).

**Appendix A. Model architecture and process representation**

[Figure]

Figure A1. Process flow diagram of the diFUME model. The detailed description of the model can be found in Stagakis et al. (2023a).

[Figure]

Figure A2. Process flow diagram of the JSBACH model. The detailed description of the model can be found in Reick et al. (2013).

[Figure]

Figure A3. Process flow diagram of the SUEWS model. The detailed description of the model can be found in Järvi et al. (2011, 2019)

[Figure]

Figure A4. Process flow diagram of the VPRM model. The detailed description of the model can be found in Mahadevan et al. (2008).

Table A1. Overview of the main equations used by each model to estimate gross primary production (GPP), ecosystem respiration (R_eco) and phenology. The original notation of each model was preserved to facilitate the comparison with the relevant publications and the flow diagrams. The parameters of the equations are explained in the Supplement. The detailed descriptions of each model can be found in the respective references provided in Section 2.3.

| | GPP | R_eco | Phenology |
|---|---|---|---|
| diFUME | $GPP = \sum_{i=1}^{n} \left[ \begin{array}{l} LAI_{sun,i}A_o(1 - exp(-aPAR_{sun,i}/A_o)) + \\ LAI_{shade,i}A_o(1 - exp(-aPAR_{shade,i}/A_o)) \end{array} \right]$ $A_o = A_{max} \cdot f(T_{air})f(VPD,\theta)$ $f(T_{air}) = exp\left[-(T_{air} - T_{opt})^2/2W^2\right]$ $f(VPD,\theta)$ based on: $g_s = \left(g_o + a_1 \frac{A_{net} f(T_{air})}{(1+\frac{VPD}{D_o})(c_s-\Gamma)}\right)\left(1 - \frac{(\theta_{ref}-\theta)^{b_1}}{(\theta_{ref}-\theta_g)^{b_1}}\right)$ | $R_{eco} = LAI\, D_{sc}R_l +$ $\lambda_{soil}R_{S,ref} exp\left[E_0\left(\frac{1}{T_{ref,S}-T_0} - \frac{1}{T_{soil}-T_0}\right)\right]\left[\frac{(\theta-\theta_0)^b}{(\theta_{ref}-\theta_0)^b}\right]$ $R_l = R_{l,ref}Q_{10}^{\frac{T_{air}-T_{ref,l}}{10}}$ | *LAI derived by Copernicus High Resolution Vegetation Phenology and Productivity (HR-VPP) product* |
| JSBACH | *Due to the complexity of the full Farquhar et al. (1980) model implementation in JSBACH, only an outline is provided here.* $GPP = LAI\, A_{stress}$ $A_{stress} = min(J_{C,stress}, J_{E,stress}) - r_d$ $g_{L,stress}^{H_2O} = f_{ws}\, g_L^{H_2O}$ | $R_{eco} = R_m + R_g + R_h$ $R_m = LAIr_d/f_{leaf}$ $R_g = (CC - 1)NPP,$ when NPP>0 $CC = (NPP + R_g)/NPP$ $R_h = \left(1 - f_{faeces}\right)F_{grazing} + k_hC_h + F_{soil}$ | $\frac{dLAI}{dt} = kLAI\left(\frac{LAI}{LAI_{max}}\right) - pLAI$ |
| SUEWS | $F_{pho} = \sum_i (fr_i F_{pho,max,i} LAI_i)g(T_{air})g(\Delta q)g(\Delta\theta)g(K_\downarrow)$ $g(T_{air}) = [(T_{air} - T_L)(T_H - T_{air})^{T_C}]/[(G_5 - T_L)(T_H - G_5)^{T_C}]$ $T_C = (T_H - G_5)/(G_5 - T_L)$ $g(\Delta q) = G_3 + (1 - G_3)G_4^{\Delta q}$ $g(\Delta\theta) = [1 - exp(G_6(\Delta\theta - \Delta\theta_{WP}))]/[1 - exp(-G_6\Delta\theta_{WP})]$ $g(K_\downarrow) = [K_\downarrow/(G_2 + K_\downarrow)]/[K_{\downarrow max}/(G_2 + K_{\downarrow max})]$ | $F_{res} = \sum_i fr_i max(a_i exp(T_{air}b_i),0.6)$ | $LAI_{d,i}$ $= \begin{cases} min(LAI_{max,i}, LAI_{d-1,i}^{\omega_{1,GDD,i}} GDD_{d,i}\omega_{2,GDD,i} + LAI_{d-1,i}), \\ \quad leaf-on, T_{base,GDD,i} < T_{d-1} \\ max(LAI_{min,i}, LAI_{d-1,i}^{\omega_{1,SDD,i}} SDD_{d,i}\omega_{2,SDD,i} + LAI_{d-1,i}), \\ \quad leaf-off, T_{d-1} < T_{base,SDD,i} \end{cases}$ |
| VPRM | $GEE = (\lambda T_{scale}W_{scale}P_{scale})EVI[1/(1 + PAR/PAR_0)]PAR$ $T_{scale} = (T - T_{min})(T - T_{max})/\left[(T - T_{min})(T - T_{max}) - (T - T_{opt})^2\right]$ $W_{scale} = 1 + LSWI/(1 + LSWI_{max})$ $P_{scale} = (1 + LSWI)/2$ | $Reco = \alpha T + \beta$ $T = T_{low}, if\, T \leq T_{low}$ | *Sentinel-2 vegetation indices* |

- Discussion of how process representation differences impact model performance across conditions

We have added a more detailed discussion focusing on how the model differences affected their performance in this study. The new text has been added in Section 4.1, which was already discussing the advantages and disadvantages of the different model types in the original manuscript. The new text is presented below:

"All the models require temperature and radiation inputs. However, VPRM requires the least number of data streams among the models (Table 2). diFUME and VPRM use satellite data to track vegetation phenology (Table A1), making them more suitable for monitoring purposes, especially in urban areas where the vegetation type heterogeneity is so pronounced that it is very difficult to model accurately. As demonstrated also by the current study, these satellite-based models are able to detect changes in leaf area due to phenological shifts or stressful events, such as drought, providing valuable insights into ecosystem responses to environmental change. However, these models are not able to predict future carbon cycling under different climate scenarios or planning strategies, as they rely heavily on observations.

Despite the different levels of sophistication of the four models on simulating GPP, all consider some sort of hyperbolic function to model the response of gross photosynthesis to light and a bell-shaped dependence on air temperature (Table A1). On the other hand, the responses of photosynthesis to vapour pressure deficit and soil water availability are accounted very differently in each model. VPRM uses a very simplistic approximation of drought effects on GPP based on LSWI, diFUME and SUEWS use empirical functions, while JSBACH has the most sophisticated approach to modelling drought effects (Section 2.3, Table A1). The generally good agreement between the models in simulated tree GPP seasonally and diurnally indicates that the simple process approximations were sufficient, possibly due to the lack of intense drought effects on tree GPP in this study based on the sap flow observations (Fig. 3a,b). On the other hand, drought effects on lawn GPP were captured by all models during August 2022, with VPRM showing the most intense drought-induced GPP reductions which were repeated in summer 2023 (Fig. 4a,b). These findings indicate that VPRM is very sensitive to LSWI and EVI indices (Fig. B1b,c), which can drive GPP to excessively low values during dry periods, in contrast to the process-based JSBACH and the empirical functions of diFUME and SUEWS. Unfortunately, we do not have any independent measurements of lawn GPP in this study to evaluate which type of model is closest to the truth.

Even though there are some similarities between the models in the representation of the GPP process, the description of $R_{eco}$ significantly differs in terms of approximations and sophistication. JSBACH is the only model tested that includes carbon pools, which are essential for studying long-term temporal dynamics. This feature allows JSBACH to model the behaviour of soil carbon pools and their changes over extended periods. For instance, if high decomposition rates persist, the decreasing soil carbon pool also decreases heterotrophic emissions, whereas other models tested do not have such feedbacks and use only empirical environmental response functions. diFUME uses a more detailed approach compared to SUEWS and VPRM, separating above and below ground respiration and using $T_{air}$, $T_{soil}$, SWC and LAI as proxies, whereas SUEWS and VPRM use an exponential and a linear response function on $T_{air}$ respectively (Table A1). Despite the differences in the representation of $R_{eco}$, the seasonal and diurnal variabilities were not very different between the four models (Fig. 3c,d, 4c,d) and the differences detected were not clearly related to the level of sophistication but rather to the choice in parametrisation. For example, diFUME tree $R_{eco}$ was higher than the other models because the parameterisation was kept the same for lawn and tree sites. JSBACH showed the best performance in predicting lawn $R_{eco}$ (Fig. 6d) but on the other hand, ecosystem process models such as JSBACH require very detailed input information (e.g. on soil carbon stocks) and are based on full-cycle assumptions that are difficult to meet in highly managed and disturbed urban ecosystems where carbon pools are constantly altered by human interventions (Golubiewski, 2006). Furthermore, process-based models such as JSBACH require a large number of input parameters compared to simple light-use-efficiency models such as VPRM which has, in some cases, been found to outperform process-based models in explaining $CO_2$ variability (Gourdji et al., 2022)."

As a key member of the SUEWS development team, I have specific concerns about the model configuration:

- Current: Near-surface measurements (2m) used to force SUEWS

- Required: Forcing data from above roughness sublayer (3-5x mean roughness element height)

- Impact: 2m measurements are inappropriate for this purpose

Thank you for raising this concern about the SUEWS modelling. We agree with your comment and have modified the forcing height to an appropriate level. Our initial idea was to use the same measurements for all the models to get a full model comparison without any biases concerning the forcing data. We now scaled the observed 2 m air temperature using a lapse rate of 6.5°C/km to height of the wind and radiation measurements (35 m) and the 2 m air temperature is simulated within SUEWS and used to calculate photosynthesis and respiration, since they strongly depend on the local conditions. The text in the manuscript (Section 2.3) was updated to include this information (see our first response above). This change had a minimal impact on the results and on the conclusions. Below you can find two figures comparing the daily aggregated GPP and Reco simulations of the original manuscript with the updated model outputs for the trees and the lawns of our study areas. We also provide in the attachment all the updated figures and tables of the manuscript using the new SUEWS outputs.

[Figure]

Fig. 1. GPP and Respiration of trees modelled with SUEWS, with previously used temperature and the updated version.

[Figure]

Fig. 2. GPP and Respiration of grass modelled with SUEWS, with previously used temperature and the updated version.

2. LAI Configuration

- Issue: Figure A1a shows modelled LAI plateau suggesting maximum LAI not properly adjusted

- Need: Clarification or correction of LAI parameterization

We thank the reviewer for this comment regarding the modelled LAI in SUEWS. The idea in the manuscript was not to adjust the model run to the local conditions but rather run SUEWS in as default mode as possible. LAI values often reach the plateau in SUEWS simulations, but actually in our case the maximum LAI is only met in a few days/week after the LAI growth. Also, we see from the comparisons that the maxima LAI values from SUEWS are in the correct order of magnitude. We did however modify the growing degree days (GDD) and senescence degree days (SDD) parameters to get more accurate spring leaf-on and autumn leaf-off (parameters provided in the Supplement). We believe that the model shows reasonable performance compared to the other models and measurements, and is comparable to previous SUEWS studies such as Omidvar et al., 2020. For clarification, we added few sentences to the SUEWS description (Section 2.3, see the text in our first response).

Omidvar, H., Sun, T., Grimmond, S., Bilesbach, D., Black, A., Chen, J., Duan, Z., Gao, Z., Iwata, H., and McFadden, J. P.: Surface Urban Energy and Water Balance Scheme (v2020a) in vegetated areas: parameter derivation and performance evaluation using FLUXNET2015 dataset, Geosci. Model Dev., 15, 3041–3078, https://doi.org/10.5194/gmd-15-3041-2022, 2022.

Technical Improvements Needed:

1. Figures and Tables

- Figure 1: Enhance map label readability

We have increased the legend fonts and some symbol sizes and colors. The new image is attached.

- Figure 2b: Correct caption misrepresenting Reco measurements

We have revised the caption text to be clearer and changed the symbol colour and size for Reco to enhance the figure readability (new figure attached). The new caption text for Fig. 2b is:

"(b) Average and standard deviation of the measured soil respiration ($R_{soil}$), measured lawn ecosystem respiration ($R_{eco}$) (sunny and shaded locations in different colour) and daily average air and soil temperatures where the shading refers to the standard deviation of the seven soil sensors."

- Table 2: Include temporal aggregation methods

The Table 2 was updated with timestep and temporal aggregation methods. The updated table is attached.

2. Other presentation improvements

- Standardise CO2 flux units throughout paper

Following the Referee's suggestion, we have kept $CO_2$ flux units uniform across the manuscript ($\mu$mol m$^{-2}$ s$^{-1}$) except for the annual totals (Fig. 5) where the units were converted to kg m$^{-2}$ a$^{-1}$ to facilitate the comparison with relevant literature.

- Define abbreviations (LSWI, EVI) at first use

EVI and LSWI are defined in the first use in Line 295 (Section 2.3.4):

"GPP is a light-dependent term using remote sensing vegetation indices, including the Enhanced Vegetation Index (EVI) and Land Surface Water Index (LSWI), combined with shortwave solar radiation to estimate the carbon uptake from photosynthesis."

- Include model parameter sets in data availability section

We have added the model parameter sets, as provided in the Supplement, in the metadata description of the datasets in Zenodo https://doi.org/10.5281/zenodo.13222637

We also made this clear in the Data availability section:

" iv. modelled CO2 fluxes and model parameter sets, are available in Zenodo repository under Creative Commons Licence CC-BY-4.0 (https://doi.org/10.5281/zenodo.13222637)"

**Response to Anonymous Referee #2**

We would like to sincerely thank Referee #2 for their time in reviewing this manuscript and for their positive feedback and valuable suggestions. Our responses to each comment are provided below in black fonts and the Referee's comments are indicated in blue fonts. All the mentioned line and section numbers refer to the originally submitted manuscript unless stated otherwise.

1. Clarify model comparison methodology (page 9): The methods section provides a detailed description of the four models used; however, a clearer explanation of the criteria and rationale for selecting each model would enhance comprehension. In other words, it would be helpful to introduce different types of models and why those four models were selected in this study.

An extra paragraph was added at the beginning of Section 2.3 to explain the rationale and criteria for selecting these four models:

"Four different types of biosphere models were selected for this study; a full carbon cycling ecosystem model designed for natural ecosystems (JSBACH), a land surface model designed for urban areas (SUEWS), a satellite-based semi-empirical $CO_2$ flux model designed for urban areas (diFUME) and a satellite-based light-use-efficiency model initially designed for natural ecosystems (VPRM). The rationale behind the model selection was to investigate if their performance in simulating the urban $CO_2$ exchanges follows the level of sophistication (i.e. the model complexity and detail in simulating the processes that govern $CO_2$ exchanges) and at the same time explore the advantages of the models that are specifically designed for urban applications over the ones designed for natural ecosystems. Furthermore, the selection of these models allows the comparison of different model attributes (e.g. satellite-based phenology versus modelled phenology), but also the identification of the challenges or drawbacks when applying models designed for natural ecosystems on urban environments. The models are described below in alphabetical order and a more detailed overview of each model can be found in the Appendix A."

2. Add justification for normalization in SWC analysis (page 14, line 380): The rationale for normalizing soil water content (SWC) in lawn areas is stated briefly. I suggest elaborating on this choice to strengthen the interpretation of temporal variability across models. Maybe some descriptions can be added into the method section.

The decision to normalize SWC values before comparing model estimates with observations is justified in more detail in the revised Section 2.5:

"The SWC observations and model estimates cannot be directly compared between them because of different soil layers considered and different soil parameterisations. Specifically, the observations were representative of approximately 9 to 21 cm soil depth, which can be closely compared to one soil layer simulated by JSBACH, but cannot be directly compared with SUEWS because the latter estimates a bulk SWC value for the total simulated soil volume. Furthermore, different soil parametrisations in the models can lead to different absolute maximum and minimum SWC values, which makes the direct comparison difficult (Fig. B1c). Therefore, the daily means of SWC observations and model estimates were normalised according to the individual time-series maximum and minimum values to avoid the effects of different model parameterisations for soil, as well as the effects of different soil depths measured and modelled. By applying this normalisation on SWC time-series, we focused on the evaluation of the temporal variability of each model rather than the absolute values."

3. Detail on seasonal effects in sap flow observations (page 15, lines 410–425): The observed sap flow seasonality was insightful. Expanding on how seasonal drought impacts this data would be beneficial for understanding tree physiology under varying moisture conditions. Also, it is interesting that why soil moisture was so stable from October to May with a high variability of precipitation?

It is indeed interesting to look more deeply into this dataset and investigate in detail the effects of drought on sap flow and transpiration for the different parks and tree species. However, this is not easily discernible in the datasets presented in this study and would require further statistical analyses looking closely on the relationships between meteorology, SWC, sap flow and LAI. Such investigation is not the main focus of this paper. A separate publication on urban tree ecophysiology using extended observation time-series and locations is planned.

The SWC does not drop during Oct – May because of sufficient precipitation and low evapotranspiration. During wintertime, the evapotranspiration rates are so low that even during the dry February the SWC does not fall significantly below the field capacity levels. The spring was fairly wet (until late May), keeping SWC close to the field capacity even though the evapotranspiration rates are expected to be high during that period. Note that the soil sensors are installed at the depth of 15 cm and the recordings are representative of approximately 9 to 21 cm soil depth. This soil layer does not show very fast responses to dry conditions which would affect the topmost soil layer.

4. Figure 4 (page 20, line 515): Consider specifying which of the Reco values in the figure pertain to sunny versus shaded locations. This adjustment would clarify the observed seasonal discrepancies across locations.

We have changed this figure and caption accordingly in order to be clearer that we present Reco observations at sunny locations separately than shaded locations. The updated figure (attached) has thicker symbols which can be more easily distinguished over the lines and the caption clearly states that sunny and shaded locations are presented separately:

"Figure 4. Time-series of model outputs for park lawns (averages of the four parks). Daily averages of (a) gross primary production (GPP), (c) ecosystem respiration ($R_{eco}$) and (e) net ecosystem exchange (NEE) estimates and diurnal hourly averages per month of (b) GPP, (d) $R_{eco}$ and (f) NEE estimates. The observations of lawn $R_{eco}$ (averages of sunny and shaded locations per campaign presented separately) are plotted over modelled $R_{eco}$ in panel (c)."

5. Addressing discrepancies in model outputs (page 19, line 490): While the manuscript discusses the general trends of model outputs, a summary or discussion of the causes behind significant discrepancies (e.g., between VPRM and diFUME) would provide a balanced perspective on model reliability.

This issue was also raised by Referee #1. We have revised the discussion Section 4.1, adding more detailed and in-depth comparisons between the model performances and discussing the possible causes behind the detected discrepancies according to the different model mechanisms (the text is provided below). We have also improved the model descriptions including process flow diagrams for each model and a Table that summarizes the key equations used by each model for the calculation of GPP, $R_{eco}$ and phenology. We have added this new information in a new Appendix and we have updated Section 2.3 (Model description) with more details, explaining better the different methodological approaches followed by each model, referring to the new information supplied in the Appendix. Additionally, we added four Tables in a Supplement document, which provide the description and the values of the parameters used by each model in this paper. For more details, please see our first three responses to the Referee #1.

New text in the discussion Section 4.1:

"All the models require temperature and radiation inputs. However, VPRM requires the least number of data streams among the models (Table 2). diFUME and VPRM use satellite data to track vegetation phenology (Table A1), making them more suitable for monitoring purposes, especially in urban areas where the vegetation type heterogeneity is so pronounced that it is very difficult to model accurately. As demonstrated also by the current

study, these satellite-based models are able to detect changes in leaf area due to phenological shifts or stressful events, such as drought, providing valuable insights into ecosystem responses to environmental change. However, these models are not able to predict future carbon cycling under different climate scenarios or planning strategies, as they rely heavily on observations.

Despite the different levels of sophistication of the four models on simulating GPP, all consider some sort of hyperbolic function to model the response of gross photosynthesis to light and a bell-shaped dependence on air temperature (Table A1). On the other hand, the responses of photosynthesis to vapour pressure deficit and soil water availability are accounted very differently in each model. VPRM uses a very simplistic approximation of drought effects on GPP based on LSWI, diFUME and SUEWS use empirical functions, while JSBACH has the most sophisticated approach to modelling drought effects (Section 2.3, Table A1). The generally good agreement between the models in simulated tree GPP seasonally and diurnally indicates that the simple process approximations were sufficient, possibly due to the lack of intense drought effects on tree GPP in this study based on the sap flow observations (Fig. 3a,b). On the other hand, drought effects on lawn GPP were captured by all models during August 2022, with VPRM showing the most intense drought-induced GPP reductions which were repeated in summer 2023 (Fig. 4a,b). These findings indicate that VPRM is very sensitive to LSWI and EVI indices (Fig. B1b,c), which can drive GPP to excessively low values during dry periods, in contrast to the process-based JSBACH and the empirical functions of diFUME and SUEWS. Unfortunately, we do not have any independent measurements of lawn GPP in this study to evaluate which type of model is closest to the truth.

Even though there are some similarities between the models in the representation of the GPP process, the description of $R_{eco}$ significantly differs in terms of approximations and sophistication. JSBACH is the only model tested that includes carbon pools, which are essential for studying long-term temporal dynamics. This feature allows JSBACH to model the behaviour of soil carbon pools and their changes over extended periods. For instance, if high decomposition rates persist, the decreasing soil carbon pool also decreases heterotrophic emissions, whereas other models tested do not have such feedbacks and use only empirical environmental response functions. diFUME uses a more detailed approach compared to SUEWS and VPRM, separating above and below ground respiration and using $T_{air}$, $T_{soil}$, SWC and LAI as proxies, whereas SUEWS and VPRM use an exponential and a linear response function on $T_{air}$ respectively (Table A1). Despite the differences in the representation of $R_{eco}$, the seasonal and diurnal variabilities were not very different between the four models (Fig. 3c,d, 4c,d) and the differences detected were not clearly related to the level of sophistication but rather to the choice in parametrisation. For example, diFUME tree $R_{eco}$ was higher than the other models because the parameterisation was kept the same for lawn and tree sites. JSBACH showed the best performance in predicting lawn $R_{eco}$ (Fig. 6d) but on the other hand, ecosystem process models such as JSBACH require very detailed input information (e.g. on soil carbon stocks) and are based on full-cycle assumptions that are difficult to meet in highly managed and disturbed urban ecosystems where carbon pools are constantly altered by human interventions (Golubiewski, 2006). Furthermore, process-based models such as JSBACH require a large number of input parameters compared to simple light-use-efficiency models such as VPRM which has, in some cases, been found to outperform process-based models in explaining $CO_2$ variability (Gourdji et al., 2022)."

6. Technical correction on figure legends (various figures): For consistency and readability, please ensure that all figure legends use uniform units (e.g., μmol $CO_2$ m$^{-2}$ s$^{-1}$ vs. g m-2 d-1) and that legends describe all variables used.

All figures and text using units g m-2 d-1 were converted to μmol $CO_2$ m$^{-2}$ s$^{-1}$. The legends were also rechecked.